# Peroxisome Proliferator-Activated Receptors and Their Novel Ligands as Candidates for the Treatment of Non-Alcoholic Fatty Liver Disease

**DOI:** 10.3390/cells9071638

**Published:** 2020-07-08

**Authors:** Anne Fougerat, Alexandra Montagner, Nicolas Loiseau, Hervé Guillou, Walter Wahli

**Affiliations:** 1Institut National de la Recherche Agronomique (INRAE), ToxAlim, UMR1331 Toulouse, France; alexandra.montagner@inserm.fr (A.M.); nicolas.loiseau@inrae.fr (N.L.); herve.guillou@inrae.fr (H.G.); 2Institut National de la Santé et de la Recherche Médicale (Inserm), Institute of Metabolic and Cardiovascular Diseases, UMR1048 Toulouse, France; 3Institute of Metabolic and Cardiovascular Diseases, University of Toulouse, UMR1048 Toulouse, France; 4Lee Kong Chian School of Medicine, Nanyang Technological University Singapore, Clinical Sciences Building, 11 Mandalay Road, Singapore 308232, Singapore; 5Center for Integrative Genomics, Université de Lausanne, Le Génopode, CH-1015 Lausanne, Switzerland

**Keywords:** peroxisome proliferator-activated receptors (PPARs), synthetic agonists, non-alcoholic fatty liver disease (NAFLD), non-alcoholic steatohepatitis (NASH), fibrosis

## Abstract

Non-alcoholic fatty liver disease (NAFLD) is a major health issue worldwide, frequently associated with obesity and type 2 diabetes. Steatosis is the initial stage of the disease, which is characterized by lipid accumulation in hepatocytes, which can progress to non-alcoholic steatohepatitis (NASH) with inflammation and various levels of fibrosis that further increase the risk of developing cirrhosis and hepatocellular carcinoma. The pathogenesis of NAFLD is influenced by interactions between genetic and environmental factors and involves several biological processes in multiple organs. No effective therapy is currently available for the treatment of NAFLD. Peroxisome proliferator-activated receptors (PPARs) are nuclear receptors that regulate many functions that are disturbed in NAFLD, including glucose and lipid metabolism, as well as inflammation. Thus, they represent relevant clinical targets for NAFLD. In this review, we describe the determinants and mechanisms underlying the pathogenesis of NAFLD, its progression and complications, as well as the current therapeutic strategies that are employed. We also focus on the complementary and distinct roles of PPAR isotypes in many biological processes and on the effects of first-generation PPAR agonists. Finally, we review novel and safe PPAR agonists with improved efficacy and their potential use in the treatment of NAFLD.

## 1. Introduction

The aim of this review is to provide information for a better understanding of the factors that impact the development, progression and complications of liver steatosis, nonalcoholic fatty liver disease (NAFLD) and nonalcoholic steatohepatitis (NASH). We discuss the roles of the nuclear receptors peroxisome proliferator-activated receptors (PPARs) in the regulation of biological processes that are participating in NAFLD, which include energy metabolism, inflammation, and fibrosis. PPARs are ligand activated transcription factors; we present new agonists that are currently in clinical trials for their potential to treat NAFLD for which no effective therapy is available.

## 2. NAFLD

### 2.1. Epidemiology

NAFLD is the most common chronic hepatic disease. It comprises a spectrum of liver conditions that can eventually lead to cirrhosis and liver cancer. Hepatic steatosis in the absence of excessive alcohol consumption is the hallmark of NAFLD, which is characterized by abnormal accumulation of triglycerides (TGs) in hepatocytes, a condition called non-alcoholic fatty liver (NAFL). The condition is considered to be benign, but can evolve into NASH, which is accompanied by hepatocyte damage and inflammation, with or without fibrosis, resulting in an increased risk of progression to cirrhosis and hepatocellular carcinoma (HCC).

The worldwide prevalence of NAFLD is constantly increasing in parallel with the global obesity pandemic. The prevalence of NAFLD is currently estimated to be approximately 25% in the general population, with the highest rates reported in South America and the USA, and the lowest in Africa [1]. A rapid and massive increase in NAFLD prevalence has also been observed in China as a result of an increase in obesity due to urbanization and lifestyle changes [2]. The trends in NAFLD incidence were followed for 17 years in a US community, finding that the incidence of NAFLD increased 5-fold, and even more (7-fold) in young adults [3]. Importantly, due to the growing increase in childhood obesity and children presenting greater vulnerability to genetic and environmental factors, NAFLD is now affecting up to 20% of the general pediatric population [4,5]. NAFLD in non-obese patients, so-called lean NAFLD, is also increasing, particularly in Asian patients [6]. Lean NAFLD is not fully understood, but possible determinants may include genetic background, different fat distribution, high fructose intake, and altered gut microbiota. Both epidemiological and preclinical studies have shown that NAFLD is more common in men than in women before menopause [7]. However, the incidence of NAFLD increases in women after menopause, suggesting a protective role of estrogens [8]. Sex-specific NASH signatures were recently identified in human liver, suggesting that NASH is a sexually dimorphic disease [9].

### 2.2. Etiology

The etiology of NAFLD is complex and involves ethnic, genetic, metabolic, and environmental factors (Figure 1).

#### 2.2.1. Ethnicity

Ethnic differences have been reported to be associated with the risk of NAFLD. For example, Hispanic individuals have a higher prevalence and severity of NAFLD [10]. Ethnic disparities are not yet well understood, but genetic and environmental factors are likely to influence the conditions associated with NAFLD, such as insulin resistance [11].

#### 2.2.2. Genetic Factors

Genome-wide association studies have identified a number of genetic factors that influence NAFLD initiation and/or progression [12,13]. The most validated genes are involved in hepatic lipid metabolism and include *PNPLA3*, *TM6SF2*, *MBOAT7,* and *GCKR*. The most common and well-described polymorphism is in *PNPLA3*. Patatin-like phospholipase domain containing protein 3 (PNPLA3) is an enzyme highly expressed in the liver that hydrolyzes TGs in hepatocytes and retinyl esters in hepatic stellate cells (HSCs). The I198M variant (rs738409, isoleucine to methionine substitution at position 148) of PNPLA3 is strongly associated with the development and progression of NAFLD [14]. This variant has decreased hydrolase activity, resulting in an accumulation of TGs and retinyl esters in lipid droplets [15,16]. At the molecular level, PNPLA3 (I148M) accumulates on lipid droplets due to defective ubiquitylation, resulting in reduced proteasome degradation [17]. In preclinical studies, overexpression of mutant PNPLA3 (I148M) in mouse liver was shown to promote hepatic TG accumulation [18]. PNPLA3 (I148M) is present at high levels in Hispanics and may represent a major determinant of ethnicity-related differences in hepatic fat accumulation [19]. However, this variant increases the risk of severe hepatic fat accumulation, inflammation, fibrosis, and HCC in different ethnicities around the world [20]. A variant of the transmembrane 6 superfamily 2 (TM6SF2) protein has also been described as a major risk factor for NAFLD [21]. TM6SF2 is predominantly expressed in hepatocytes and enterocytes and localized in the endoplasmic reticulum and Golgi. The protein’s exact function remains elusive, but it may be involved in very low-density lipoprotein (VLDL) formation in hepatocytes. The E167K (rs58542926, glutamate to lysine substitution at postion 167) loss of function variant of the protein causes higher liver fat content and fibrosis, but reduced secretion of VLDL and serum TGs [22,23,24]. This variant is also associated with reduced cardiovascular risk due to lower levels of circulating VLDL [25]. Results obtained from mouse studies, though often controversial, clearly indicate that the level of TM6SF2 protein is an important determinant of lipoprotein metabolism and NAFLD [26]. More recently, a polymorphism (rs641738) in the locus carrying the membrane bound O-acyltransferase domain-containing 7 (MBOAT7) gene has been associated with the risk and severity of NAFLD [27]. MBOAT7, also known as lysophosphatidylinositol acyltransferase (LPIAT1) is an enzyme involved in hepatic phospholipid remodeling by transferring polyunsaturated fatty acids to lysophospholipids. The variant rs641738 results in suppression of MBOAT7 at the messenger RNA and protein levels, altered phosphatidylinositol profiles, and was recently associated not only with steatosis development, but with more severe liver damage and advanced stages of fibrosis [28], as well as HCC in patients without cirrhosis [29]. In mice, downregulation of MBOAT7 leads to hepatic steatosis associated with obesity [30]. The rs1260326 polymorphism in the glucokinase regulator (GCKR) gene is a loss-of-function mutation that has also been linked to NAFLD development [31]. GCKR negatively regulates glucokinase in response to fructose-1-phosphate, modulating glucose uptake in the liver. The rs1260326 variant results in increased hepatic glucose uptake and malonyl-CoA concentration, providing more substrates for de novo lipogenesis [32]. This GCKR variant is also highly associated with fatty liver in obese youths [33]. Recent genetic and epidemiological studies have identified other polymorphisms associated with NAFLD progression in several genes involved in retinol metabolism (hydroxysteroid 17-beta dehydrogenase 3 (*HSD17B3*)), glycogen synthesis (protein phosphatase 1 regulatory subunit 3B (*PPP1R3B*)), bile acid homeostasis (beta-klotho (*KLB*)), oxidative stress (uncoupling protein 2 (*UCP2*)*,* superoxide dismutase 2 (*SOD2*)), insulin signaling pathway (tribbles pseudokinase 1 (*TRIB1*)), and inflammation (suppressor of cytokine signaling 1 (*SOCS1*), interferon lambda 3 (*IFNL3*)*,* MER proto-oncogene tyrosine kinase (*MERTK*)) [12,34,35,36,37,38,39]. In addition, epigenetic mechanisms, including post-translational histone modifications, DNA methylation, and micro-RNAs, are important in disease development [12]. A recent review presented a new prediction model that describes enriched genetic pathways in NAFLD, defined as the NAFLD-reactome [40]. Yet another layer of complexity has emerged with several genetic polymorphisms associated with both NAFLD and other liver diseases and metabolic disorders [41].

#### 2.2.3. Metabolic Factors

In addition to ethnic and genetic factors, several metabolic and environmental factors contribute to NAFLD (Figure 1). Metabolic syndrome is defined as the presence of three of the five following conditions: high serum TGs, low serum high-density lipoprotein (HDL), elevated systemic blood pressure, hyperglycemia, and central obesity. Metabolic syndrome is recognized as a strong risk factor of NAFLD development and progression [42]. In a large cohort study including different ethnic groups, the prevalence of NAFLD increased in subjects with more metabolic syndrome criteria, reaching 98% when all five criteria were present [43]. NAFLD is not only associated with metabolic syndrome in general, but also with its individual conditions. Among NAFLD patients, the prevalence of metabolic syndrome is 42%, obesity 51%, type 2 diabetes mellitus (T2DM) 22%, dyslipidemia 69%, and hypertension 39% [1]. In patients with T2DM and normal circulating aminotransferase levels, the prevalence of NAFLD has been estimated to be 50% [44]. A recent meta-analysis including more than 35000 T2DM patients reported a pooled prevalence (24 studies) of NAFLD of 60% [45]. Based on histopathological assessment, T2DM patients also have a high risk of developing NASH and advanced fibrosis [46,47]. Obesity has been identified as an independent risk factor, with a 3.5-fold increased risk of developing NAFLD [48], and a linear relationship exists between body mass index (BMI) and NAFLD/NASH prevalence [49]. Several studies have highlighted the importance of fat distribution in showing that the amount of visceral fat is higher in NAFLD patients [50] and correlates with the severity of the disease [51], whereas large subcutaneous fat areas are associated with regression of NAFLD. These findings suggest that different types of body fat can increase or reduce the risk of NAFLD [52]. Several dyslipidemia phenotypes have been described in NAFLD patients [53], and are characterized by an increase in small dense low density lipoprotein (LDL) particles [54], higher postprandial lipemia after an oral fat meal [55], and HDL dysfunction [56]. Several studies have also reported that hypertension increases the risk of NAFLD [57] and the risk of NAFLD progression to fibrosis [58].

#### 2.2.4. Environmental Factors

Environmental factors, especially dietary factors, also contribute to NAFLD development and progression [59,60]. The Western diet, which is particularly rich in added fructose, is associated with a greater risk of NAFLD, whereas the Mediterranean diet, which is high in polyunsaturated fatty acids (PUFAs), monounsaturated fatty acids (MUFAs), and fiber, has a beneficial effect on NAFLD [61,62]. Several nutrients impact the metabolic pathways leading to the lipid accumulation that characterizes the NAFLD initiation step, whereas others modulate key features in the pathogenesis of NASH, such as oxidative stress and inflammation.

NAFLD patients have been shown to have higher fructose intake due to sweetened beverage consumption [63], which is associated with the progression of fibrosis and inflammation [64]. Fructose consumption has dramatically increased in the last few decades, in parallel with the increased use of added sugars in the form of sucrose and high-fructose corn syrup in processed foods and beverages [65]. Mechanistically, fructose stimulates de novo lipogenesis, a central mechanism of hepatic lipid accumulation in NAFLD (see Section 2.3). Fructose metabolism rapidly induces precursors of lipogenesis, leads to ATP depletion, the suppression of mitochondrial fatty acid oxidation, and the production of carbohydrate metabolites, which activate the lipogenesis transcriptional program via the transcription factors carbohydrate-responsive element-binding protein (ChREBP) and sterol regulatory element binding protein 1c (SREBP1c) [65,66]. Recently, a novel pathway of lipogenesis activation by fructose, in which fructose is converted to acetate by the gut microbiota, was described. This pathway results in lipogenic pools of acetyl-CoA [67]. Fructose metabolism also leads to uric acid production, which has pro-oxidative and pro-inflammatory effects [68].

High-fat diets (HFDs) induce obesity and insulin resistance, which are strongly associated with NAFLD. A meta-analysis including 1400 NAFLD patients suggested that omega-3 PUFA supplementation has a beneficial effect on liver fat [69]. In contrast, no effect of omega-3 supplementation on NASH has been reported [70]. Omega-3 PUFAs, which are particularly abundant in fish oil, impact the activity of transcription factors, such as PPARα [71], liver X receptor (LXR) [72], ChREBP [73], SREBP1c [74], and peroxisome proliferator-activated receptor-gamma coactivator 1β (PGC1β) [75], which control the expression of genes involved in fatty acid homeostasis [76]. In a cohort of T2DM patients, a MUFA-rich diet induced a reduction in liver fat content [77], potentially through an increase in hepatic beta-oxidation [78]. In preclinical studies, dietary cholesterol has been shown to promote NASH and fibrosis, and contribute to HCC progression [79,80]. In human studies, high cholesterol levels have been associated mostly with cirrhosis and liver cancer [81].

High-protein diets have prevented hepatic lipid accumulation in animal studies [82,83,84]. In a small cohort of healthy men, a high-protein diet rich in glutamate increased plasma short-chain TG levels, which was interpreted as having resulted from increased de novo lipogenesis [85]. In contrast, alterations in plasma amino acid concentrations are clearly associated with the occurrence and severity of NAFLD [86], especially amino acids that are involved in glutathione synthesis, such as glycine, serine, and glutamate [87,88]. The current literature also suggests that branched chain amino acids (BCAAs) are increased in the plasma of NAFLD patients [89,90]. Interestingly, plasma BCAA levels correlate with NAFLD severity in a sex-dependent manner, increasing with disease severity in women, but decreasing in men [91]. Increased BCAA levels may be due to impaired BCAA catabolism by the gut microbiota [90]. Preclinical studies have proposed that BCAAs promote steatosis by increasing adipocyte lipolysis and decreasing the conversion of free fatty acids (FFAs) into TGs [92]. Micronutrients, such as vitamins, also play an important role in NAFLD. Plasma vitamin D levels are inversely associated with the severity of NAFLD [93]. In adult patients with NAFLD, vitamin E supplementation improves steatosis and hepatic inflammation, but has no effect on fibrosis [94]. Animal studies suggest that vitamin E ameliorates NAFLD/NASH by attenuating oxidative stress and inflammation [95].

Increasing epidemiological and experimental evidence suggests that exposure to some environmental contaminants could contribute to NAFLD progression [96,97,98,99]. Pesticides, insecticides, fungicides, and herbicides have demonstrated hepatotoxic effects by modulating lipid metabolism, inflammation, and oxidative stress [100].

#### 2.2.5. Gut Microbiota

In recent years, gut microbiota and microbiota-derived compounds have emerged as important players in the pathogenesis of NAFLD in mice and humans [101,102]. Gut microbiota have been shown to cause NAFLD in animal studies. In humans, NAFLD severity is associated with gut dysbiosis, with an enrichment of Bacteroides in NASH patients compared to matched healthy individuals [103]. A recent review described the microbiome signature in human NAFLD according to the different stages of disease severity [104]. Suggested mechanisms by which the gut microbiota impact NAFLD and its progression include increased intestinal permeability [105], leading to the release of bacterial endotoxins (lipopolysaccharide (LPS)), and microbiota-derived factors (short-chain fatty acids), which may trigger inflammatory responses and affect hepatic metabolism via the modulation of metabolic gene expression [106]. As mentioned above, the gut microbiota converts fructose into acetate, which fuels hepatic lipogenesis [67]. Human NAFLD studies have some limiting factors that have not always been considered, such as possible confounding effects of obesity, insulin resistance, and T2DM on dysbiosis, as well as the variable demographic characteristics of the analyzed cohorts. Together with the use of different sequencing tools and NAFLD diagnostic methods, they may have been responsible for the discrepancy observed in microbiome signatures [104].

### 2.3. Pathophysiology

The pathogenesis of NAFLD and its complications are complex and not fully understood. As described previously, several factors acting in collaboration or synergy contribute to NAFLD development and its progression to NASH, leading to the multiple parallel hit hypothesis of NAFLD progression [107] (Figure 2).

Hepatic steatosis is characterized by excessive accumulation of TGs in hepatocytes due to an imbalance between FFA influx and export, and/or catabolism. Increased FFAs within the livers of NAFLD patients originate primarily from adipose lipolysis (59%), followed by de novo lipogenesis (26%) and diet (15%) [108,109]. Both adipose lipolysis and de novo lipogenesis are normally regulated by insulin. However, NAFLD patients are usually insulin-resistant, and insulin is not able to suppress lipolysis, leading to increased circulating FFAs arriving to the liver. Adipose tissue contributes to NAFLD by modulating the lipid flux to the liver and via production of hormones and cytokines that impact hepatocyte physiology [110,111]. In the liver, insulin also fails to inhibit hepatic glucose production, but continues to stimulate lipid synthesis, leading to hyperglycemia, hyperlipidemia, and steatosis. This paradox of hepatic insulin resistance, which is also associated with obesity and T2DM, is still not fully understood. Current hypotheses to explain the selective hepatic insulin resistance involve extrahepatic pathways from peripheral metabolic organs [112], which underscores the multi-organ dimension of NAFLD pathogenesis. Insulin stimulates lipogenesis through the transcription factor SREBP1c, which regulates the expression of genes encoding enzymes involved in de novo lipogenesis [113,114]. The resulting hyperglycemia activates the glucose-responsive transcription factor ChREBP, which is also an important regulator of lipogenic gene expression [115,116]. Both SREBP1c and ChREBP are required for the maximal postprandial enhancement of lipogenesis [117]. The lipogenesis product malonyl-CoA inhibits fatty acids from associating with carnitine by down-regulating the enzyme carnitine acyltransferase, which reduces their entry into mitochondria and their beta-oxidation, thereby contributing to the overall increase in hepatic lipids. Adipose insulin resistance also leads to adipose tissue defects, including decreased secretion of adiponectin, an adipokine that increases beta-oxidation and decreases de novo lipogenesis in the liver [118]. When both fatty acid catabolism and export via VLDL secretion are not sufficient to compensate for the hepatic lipid overload, toxic fatty acid derivatives are produced that promote steatosis progression to NASH [119]. NASH is characterized by fat deposition, inflammation, ballooned hepatocytes, hepatocyte apoptosis and necrosis, and a variable rate of fibrotic progression. In hepatocytes, candidate lipotoxic lipids include saturated fatty acids, lysophosphatidylcholine, ceramides, sphingolipids, and diacylglycerol [120]. Hepatic free cholesterol levels are also elevated in NASH patients and contribute to liver toxicity [121]. A specific lipid signature that discriminates between control, steatotic, and NASH patients has been established and highlights dysregulation in the long-chain fatty acid (LCFA) synthesis pathway in NASH, leading to accumulation of LCFA and a decrease in phospholipids [122]. In response to lipid-induced hepatocellular injury, inflammasomes become activated, and endoplasmic reticulum (ER) and oxidative stress increase, leading to pro-inflammatory cytokine production, lipid peroxidation, hepatocyte cell death (apoptosis and necrosis), and aggravated liver damage. Chronic hepatocyte injury induces the recruitment and Toll-like receptor (TLR)-dependent activation of inflammatory cells, mainly liver macrophages or Kupffer cells, which amplifies inflammation and apoptosis. Kupffer cells also produce activating factors (platelet-derived growth factor [PDGF] and transforming growth factor β [TGFβ]) for the activation of HSCs, which proliferate and secrete collagen, as well as other extracellular matrix proteins, leading to fibrosis [123].

### 2.4. Progression and Associated Diseases

NAFLD progression is still not clearly understood due, in part, to its heterogeneity. Data indicate that all NAFLD patients have a risk of developing progressive liver disease over time. However, fibrosis is currently the best histopathological predictor of hepatic complications and disease-related mortality [124,125], and stage 2–4 fibrosis is predictive of cirrhosis-related issues [126]. In general, NAFLD is a slowly progressive disease, and many patients will develop cirrhosis or liver-related mortality; among NAFL patients who are considered to suffer from a benign condition, approximately 25% may progress to liver fibrosis. Identifying these patients and providing effective treatment remains a challenge [124]. Other patients will develop NASH, and these patients are more prone to progress to advanced stages of the disease. Overall, it means that some patients will remain at a stable steatosis stage, some will progress to NASH with or without fibrosis, and others will develop fibrosis without NASH (Figure 2). Using paired biopsies, McPherson et al. reported that 44% of patients with NAFL developed NASH, but that fibrosis progression was not different between patients with NASH and patients with NAFL at baseline [127]. A meta-analysis of paired liver biopsy studies in patients with NAFLD confirmed that fibrosis progression does not differ between NAFL and NASH patients, with an overall 35–40% of patients developing fibrosis [58]. Compared to matched controls, NAFLD patients are at higher risk of HCC [128], and the incidence of HCC was higher in NAFLD patients with cirrhosis than in those without cirrhosis [129]. Collectively, these data confirm the heterogeneous nature of NAFLD and led to classifying patients as fast and slow progressors. Slow progressors may develop NASH but have a low risk of fibrosis, whereas fast progressors rapidly progress from steatosis to advanced fibrosis [130].

Due to its culmination in cirrhosis and HCC, NAFLD is becoming the major cause of liver transplantation. In addition to liver-related complications, NAFLD is also highly associated with an increased risk of extra-hepatic cancer [131], as well as cardiovascular and metabolic diseases. As described above, T2DM, hypertension, and cardiovascular disease (CVD) are major risk factors for NAFLD, but the link between these cardiometabolic diseases and NAFLD is more complex than initially thought. Clinical and experimental evidence now suggests a bi-directional relationship and indicate that NAFLD may precede and promote T2DM, hypertension, and CVD, rather than being the result of these conditions [132]. The incidence of metabolic comorbidities, cardiovascular events, and mortality was studied in a cohort of NAFLD patients followed for 17 years [3]. Patients with NAFLD had more diabetes, hypertension, and hyperlipidemia, increased risk of cardiovascular events and mortality, and shorter life expectancy than patients without NAFLD.

Altogether, NAFLD is a complex, multi-factorial, metabolic disease, the development and progression of which are strongly influenced by ethnicity, genetic predisposition, and metabolic and environmental risk factors (Figure 1). In addition, interactions between all of these factors, especially gene-diet interaction, promote NAFLD development, which has boosted the emergence of nutrigenomics as a novel approach for the management of NAFLD patients [133]. The pathogenesis of NAFLD is complex and involves many hepatic mechanisms, such as defects in lipid and glucose metabolism and insulin resistance, and important cross-talk between the liver and other organs in the adipose-liver and gut-liver axes, including important roles of the microbiota (Figure 2). Moreover, in contrast to NAFL, which can easily be detected by ultrasound and plasma biochemistry, the diagnosis of NASH and fibrosis requires liver biopsy for precise staging, which remains a limitation for the diagnosis of advanced phases of the disease. Despite many drugs being in development, there is currently no U.S. Food and Drug Administration (FDA)-approved pharmacological therapy for NAFLD treatment.

## 3. Current Therapeutic Strategies for NAFLD

### 3.1. Lifestyle Modification and Bariatric Surgery

NAFLD is considered the hepatic expression of metabolic syndrome and is closely associated with morbidities, such as obesity and insulin-resistance. Thus, weight loss represents the primary effective strategy for NAFLD management. Weight loss can be achieved through different interventions, including lifestyle changes, pharmacotherapy, and surgical procedures, and improves NAFLD biomarkers, though it effect on liver fibrosis is not significant [134]. In the absence of an approved drug therapy for NAFLD/NASH, weight loss through lifestyle interventions (exercise, diet) remains the first-line treatment. Bariatric surgery, which can be performed using minimally invasive techniques, also represents an effective option.

#### 3.1.1. Exercise

Aerobic exercise refers to physical exercise usually performed at light-to-moderate intensity over a relatively long period, during which increased breathing brings oxygen into the body to sustain aerobic metabolism. Eight weeks of aerobic exercise in different forms reduces hepatic fat independently from the dose and intensity of the exercise [135]. Liver fat content is also reduced in pre-diabetic patients with NAFLD who are subjected to Nordic walking for 8 months [136]. These results are supported by a recent meta-analysis that found that exercise training alone has a beneficial effect on liver fat content, even in the absence of significant weight loss [137].

High-intensity interval training (HIIT), which alternates short periods of intense exercise with less intense recovery periods, performed three times per week for 12 weeks has been reported to reduce liver fat by 27% in adult NAFLD patients compared to individuals on standard care, and to also improve cardiac function [138]. Eight-week HIIT also has a beneficial effect on intra-hepatic TGs in obese diabetic patients with NAFLD [139]. Twelve weeks of HIIT reduces inflammatory markers and improves hepatic stiffness in obese men with NAFLD, suggesting that HIIT may have beneficial effects in patients with NASH [140]. Collectively, these data show that HIIT regimens significantly reduce hepatic fat in NAFLD/NASH patients.

Resistance training is a form of physical activity that causes muscle contraction against an external resistance and improves strength and endurance. Resistance exercise for 8 weeks reduces hepatic lipids in NAFLD patients [141], and 3 months of resistance training reduces liver fat content in NAFLD patients, but without a significant change in weight [142]. Interestingly, combined aerobic and resistance training improves aerobic capacity and skeletal muscle strength, and may be the most effective exercise program for improving NAFLD [143].

Collectively, exercise in whatever form appears to reduce the liver fat content, even in the absence of weight loss. No significant difference has been found between aerobic or resistance training in the reduction of liver fat, whereas continuous training of moderate volume and moderate intensity seems to be more beneficial [144,145]. Interestingly, although combining an exercise program with dietary interventions augments the reduction in hepatic fat content, exercise only is also effective in reducing hepatic lipid content in NAFLD patients [136,146]. As most of the studies have been performed with diabetic and/or obese NAFLD patients, the beneficial effect of exercise still needs confirmation in large-scale prospective studies, as a recent meta-analysis showed that physical activity only slightly reduces liver fat content in non-diabetic NAFLD patients [147]. Interestingly, starting to exercise has been independently associated with NAFLD remission only in men, suggesting a sex-specific hepatic response to exercise [148].

The above studies are informative, but the mechanisms underlying the reduction in hepatic fat following exercise are poorly studied. Proposed mechanisms of action include changes in liver physiology, such as increased VLDL clearance and improved mitochondrial fatty acid oxidation, together with extra-hepatic effects, such as improved peripheral insulin sensitivity, decreased visceral fat, and improved cardiovascular function [149,150].

#### 3.1.2. Dietary Interventions

Dietary modifications remain the most effective physiological intervention for losing weight. Therefore, several studies have analyzed the effects of different dietary patterns on NAFLD development and progression. Currently, the Mediterranean diet is recommended for the management of NAFLD [151]. The Mediterranean diet has been shown by proton magnetic resonance spectroscopy to reduces liver steatosis in obese NAFLD patients without changes in body weight [152]. Adherence to the Mediterranean diet reduces the severity of liver disease among NAFLD patients and is associated with lower insulin resistance [153]. These findings are supported by two recent systematic reviews, which reported a reduction in hepatic steatosis in patients with NAFLD following the Mediterranean diet [154,155].

Caloric restriction leading to weight loss has also been associated with improved metabolic parameters in patients with NAFLD. A 12-month hypocaloric diet improved NASH-related histological parameters (steatosis, inflammation, and ballooning) in a paired biopsy study. In addition, individuals with weight loss > 10% have better NASH resolution and present with a regression of fibrosis, reinforcing the importance of weight loss in NAFLD management [156].

Given the detrimental hepatic effects of carbohydrates, especially fructose as described above, very low-carbohydrate ketogenic diets have received attention for the management of NAFLD. However, though ketogenic diets have largely been analyzed in rodents, only a few studies have been performed in humans. A pilot study in obesity-associated fatty liver disease showed that patients on a 6-month ketogenic diet lost weight and presented with histological improvements in steatosis, inflammation, and fibrosis [157]. Recently, a short-term ketogenic diet was shown to decrease hepatic lipids in obese patients in only 6 days, despite increased plasma FFA levels. This effect is attributed to an increase in hepatic TG hydrolysis and the use of released fatty acids for ketogenesis [158]. Another recent study reported that 1 year on a carbohydrate-restricted diet reduces the risk of fatty liver and advanced fibrosis in obese diabetic patients [159]. Notably, these two studies included obese and/or diabetic patients with suspicion of NAFLD, but imaging- or biopsy-proven NAFLD was not documented.

High-protein foods for weight loss have received much attention in recent years, and have started to be tested in NAFLD patients, but still remain poorly studied. A 2-week isocaloric, low-carbohydrate diet with increased protein content promotes multiple metabolic benefits in obese NAFLD patients, including a reduction in hepatic lipids due to decreased de novo lipogenesis and increased beta-oxidation. Interestingly, these changes are associated with an alteration in the composition of the gut microbiota [160]. Ketone bodies produced in response to carbohydrate restriction can induce additional protective effects in NAFLD, such as anti-oxidant and anti-inflammatory effects [161]. Another recent study analyzed the effects of isocaloric diets rich in animal proteins or plant proteins for 6 weeks in diabetic patients with NAFLD and found that both high-protein diets reduce liver fat [162]. As several studies have highlighted the role of the gut microbiota in NAFLD pathogenesis, supplementation with probiotics has been tested in NAFLD patients. To date, clinical data from such studies are disputed, but most of them report loss of body weight, suggesting that probiotic supplementation can be used as a complementary approach for patients with NAFLD [163]. Furthermore, high intake of insoluble dietary fiber correlates with a lower prevalence of NAFLD [164], and high-fiber diets promote short-chain fatty acid producing microbiota with beneficial effects in T2DM patients [165]. Clinical data on such diets in NAFLD patients are currently lacking. Recently, 1-year administration of a symbiotic combination (one probiotic and one prebiotic) was shown to change the fecal microbiome but had no effect on liver fat or fibrosis compared to placebo in NAFLD patients [166].

Taken together, observations from dietary interventions show that the Mediterranean diet and caloric restriction are beneficial for patients with NAFLD. As mentioned above, the macronutrient composition of the diet also appears to be important; saturated fatty acids and simple sugars damage the liver, whereas MUFAs, PUFAs, and dietary fiber induce beneficial hepatic effects [167]. More recently, studies have underscored that meal timing and frequency may also be important [143]. Studies in rodents have suggested that intermittent fasting and restricted feeding can have beneficial effects on NAFLD, and the few human studies agree that regular meals combined with regular fasting periods may provide physiological benefits (inflammation, circadian rhythm, autophagy, stress resistance, and gut microbiota) [168]. Combined diet and exercise interventions may induce greater benefits, though the current data are controversial [150]. Moreover, though lifestyle interventions (diet and exercise) are effective in reducing intrahepatic lipids without changes in body weight, weight loss appears to be required for improvement in NASH and fibrosis. Interestingly, lessening of NAFLD was measured in 67% of non-obese patients following lifestyle intervention [169].

The susceptibility to developing NAFLD comprises inherited risk factors, as described earlier, such as I148M PNPLA3, E167K TM6SF2, P446L GCKR, and rs641738 in *MBOAT7*. As these variants are nutrient-sensing, nutritional genomics approaches can be utilized in the future as interventions that make use of beneficial nutrients suitable to the patients’ genomes and avoid those that have unhealthy effects. This avenue remains to be explored, though several ongoing clinical trials are already testing nutrigenomic diets in NAFLD patients [170].

#### 3.1.3. Bariatric Surgery

Bariatric surgery is another effective non-pharmacological weight-loss therapy, and is indicated for patients with a BMI > 35 and severe comorbidities, such as T2DM and hypertension. Several studies have reported resolution of steatosis, as well as NASH and fibrosis, in patients who have undergone weight-loss surgery [171,172]. According to a meta-analysis of 21 studies, bariatric surgery results in histological or biochemical improvement of steatosis, NASH, and fibrosis in 88%, 59%, and 30% of NAFLD patients, respectively [173]. Furthermore, patients with NAFLD who undergo bariatric surgery have a lower risk of progression to cirrhosis compared to matched controls without surgery [174]. Bariatric surgery has beneficial effects through both weight loss and effects on metabolic pathways involved in NAFLD, including improved glucose and lipid homeostasis and decreased inflammation [175]. In a prospective study evaluating fibrosis and NASH in severely obese patients, most of the patients had low levels of NAFLD 5 years after surgery, but fibrosis had slightly increased [176]. Overall, bariatric surgery is very effective for reducing weight, but its effect on fibrosis progression is not yet clear and requires further attention. In addition, complications associated with this invasive procedure, such as sepsis and hemorrhage, limit its application.

In conclusion, lifestyle interventions and bariatric surgery are effective in NAFLD, especially through the induction of weight loss. However, studies are still needed to clarify the long-term effect of these interventions.

### 3.2. Pharmacotherapy

For most patients, lifestyle interventions such as diet and exercise, although effective, are difficult to achieve, and even more difficult to maintain. Thus, the development of pharmacological treatments is necessary. Most of the current pharmacological interventions aim at decreasing metabolic risk factors, such as obesity, insulin resistance, dyslipidemia, and hypertension. A systemic review of 29 randomized controlled trials testing several anti-diabetic drugs in NAFLD patients with and without T2DM reported that all anti-hyperglycemic agents have beneficial effects, at least on serum liver enzymes [177]. Among these anti-diabetic agents, pioglitazone is recommended for NAFLD patients with T2DM [178]. Vitamin E, which has anti-oxidant activity, is another current strategy for NASH management in patients without T2DM.

#### 3.2.1. Pioglitazone

Pioglitazone is a thiazolidinedione that improves insulin resistance and glucose and lipid metabolism in T2DM. The phase 3 Pioglitazone vs Vitamin E vs Placebo for Treatment of Non-Diabetic Patients With Nonalcoholic Steatohepatitis (PIVENS) trial examined the effect of pioglitazone and vitamin E in non-diabetic patients with biopsy-proven NASH after 96 weeks of treatment. Compared to placebo, pioglitazone was associated with reduced hepatic steatosis, inflammation, and ballooning, but it did not improve fibrosis [94,179]. Several other studies have reported that pioglitazone treatment leads to histological improvement of steatosis and inflammation in subjects with NASH from 6 months of treatment onwards [180,181]. Interestingly, a retrospective analysis of data collected from the PIVENS trial suggested a strong link between the histological features of NASH resolution and improved fibrosis in NASH [182]. Nevertheless, the benefit of pioglitazone on fibrosis remains to be clarified because of divergent results. Some studies have reported an improvement in fibrosis [181,183,184], whereas others have reported no change in fibrosis [94,180].

In a recent study, patients with biopsy-proven NASH and prediabetes or T2DM were given pioglitazone or placebo for 18 months. A reduction in intrahepatic TG content and NASH resolution was observed in both groups, whereas fibrosis was reduced only in the T2DM patients [185]. Interestingly, genetic factors could contribute to the variability in the response to pioglitazone in NASH patients [186]. Adverse effects of pioglitazone include body weight gain, fluid retention, bone loss, and heart failure [187]. Furthermore, prediabetic and diabetic NASH patients treated with pioglitazone for 3 years exhibit decreased bone mineral density at the level of the spine, which is already present after 18 months of treatment compared to placebo [188]. A systematic review and meta-analysis concluded the risk of bladder cancer may be increased by pioglitazone and, therefore, recommend that patients on high-dose and long-term pioglitazone treatment be examined regularly for manifestations of bladder cancer [189]. Another systemic review of observational studies of the association between pioglitazone use and bladder cancer concluded that further research needs be conducted to clarify the role of pioglitazone use in the incidence of this cancer [190].

Preclinical studies have greatly contributed to our understanding of the mechanisms underlying the beneficial effects of pioglitazone. Pioglitazone is a ligand of PPARγ, a member of the nuclear receptor superfamily that is highly expressed in adipose tissue and plays a key role in glucose regulation and lipid metabolism [191]. Hepatoprotective effects of pioglitazone include increased insulin sensitivity, adipose TG storage, and adiponectin production, as well as decreased pro-inflammatory cytokine production by adipose tissue and macrophages [192,193]. These effects lead to a reduction in fatty acid delivery to the liver and decreased inflammation. In a murine model of NASH (high fructose and high trans fat), pioglitazone improves the toxic lipid profile by increasing the hepatic mitochondrial oxidative capacity and changing whole body BCAA metabolism [194]. Pioglitazone reduces HFD-induced steatosis in mice by stimulating the hepatic expression of genes and proteins involved in lipolysis, beta-oxidation, and autophagy [195]. In adiponectin-deficient mice, the reduction of HFD-induced steatosis by pioglitazone is blunted, revealing a role of adiponectin in this process [196].

#### 3.2.2. Vitamin E

Vitamin E, which is known for its anti-oxidant effects, is considered the first-line treatment in NAFLD patients without T2DM. The PIVENS trial showed that vitamin E improves NASH compared to placebo (43% vs. 19%) in NAFLD patients without diabetes. As for pioglitazone, there was no improvement of fibrosis after 96 weeks of treatment [94]. Resolution of NASH in this cohort correlated with increased HDL levels, decreased TG levels, and reduced lipoprotein-related CVD risk compared to patients without an improvement in NASH [197,198]. The effect of vitamin E on NASH resolution was confirmed in non-diabetic children with NASH in the Treatment of Nonalcoholic Fatty Liver Disease in Children (TONIC) trial despite no improvement in liver enzyme levels [199]. Interestingly, the vitamin E response in non-diabetic NASH patients has been linked to the genotype of haptoglobin (Hp), an anti-oxidant protein that prevents hemoglobin-mediated oxidative injury. Two alleles of Hp (Hp 1 and Hp 2) generate three distinct genotypes (Hp 1-1, Hp 2-1, and Hp 2-2). NASH patients carrying at least one Hp 2 allele respond better to vitamin E treatment in terms of steatohepatitis resolution, histological improvement, and NAFLD activity score (NAS) compared to those with the Hp 1-1 genotype [200]. In contrast, in diabetic patients with biopsy-proven NASH, vitamin E supplementation for 18 months did not significantly reduce the NAS compared to placebo, despite resolution of NASH in 42% of patients vs. 18% with placebo. In this study, the effects of a combination of vitamin E and pioglitazone on liver histology were also examined. Though no change in fibrosis was observed, steatosis, inflammation, and ballooning were reduced by the combination therapy [201]. However, whether the combination of vitamin E and pioglitazone is more beneficial than pioglitazone alone was not examined. Others have seen differences regarding the vitamin E response between diabetic and non-diabetic individuals. The serum vitamin E concentration is higher in diabetic NAFLD patients, and there is an inverse relationship between vitamin E levels and all-cause mortality only in NAFLD patients without diabetes [202]. Clinical use of vitamin E has been limited because its long-term treatment has been associated with prostate cancer [203] and hemorrhagic stroke [204]. Vitamin E may also increase the risk of overall mortality, though this remains controversial [205,206].

The mechanisms of vitamin E action have been investigated in several rodent models of NAFLD. Well known for its anti-oxidant activities, vitamin E contributes to the scavenging of reactive oxygen species (ROS) and reactive nitrogen species (RNS), increase in the anti-oxidative enzyme superoxide dismutase (SOD), and inhibition of lipid peroxidation [207]. Recently, vitamin E supplementation for 2 weeks in HFD-fed mice showed beneficial effects on lipid accumulation and glucose homeostasis through activation of the transcription factor nuclear factor erythroid-2-related factor 2 (Nrf2) and upregulation of carboxylesterase 1 (CES1) [208]. In addition, vitamin E reduces apoptosis and inflammation through regulation of M1/M2 macrophage polarization and inhibition of T-cell recruitment [95]. Moreover, vitamin E induces adiponectin expression via a PPARγ-dependent mechanism [209].

In summary, the two classic therapies vitamin E and pioglitazone have beneficial effects on steatosis and inflammation. Vitamin E does not improve liver fibrosis, which is the strongest indicator of mortality in NAFLD patients, and the effect of pioglitazone on fibrosis varies from study to study. Furthermore, adverse effects and uncertain long-term benefits associated with both pioglitazone and vitamin E have limited their clinical use in NAFLD.

#### 3.2.3. Other Current and Emerging Medications

Several other known molecules have been investigated or are currently under investigation in clinical trials for their effectiveness in NASH patients. Most of these medications target metabolic comorbidities and have been approved for the treatment of other diseases closely associated with NAFLD, such as obesity, dyslipidemia, and T2DM. For example, orlistat is an intestinal lipase inhibitor indicated for the treatment of obesity; statins are inhibitors of the enzyme hydroxymethylglutaryl-coenzyme A (HMG-CoA) reductase that are used to treat dyslipidemia due to their lipid-lowering effect; glucagon-like peptide (GLP-1) receptor agonists and dipeptidyl peptidase-4 (DPP-4) inhibitors increase incretins and are approved for the treatment of diabetic patients. In addition, the GLP-1 receptor agonist, liraglutide, is being investigated in a phase 2 clinical trial in NASH patients, the Liraglutide Efficacy and Action in NASH (LEAN) study [210]. All of these medications are effective in reducing hepatic steatosis, but no changes in liver inflammation or fibrosis have been reported [211]. As NAFLD is characterized by a disturbance in lipid and glucose homeostasis, drugs targeting de novo lipogenesis and glucose metabolism, such as stearoyl-CoA desaturase 1 (SCD1) and acetyl CoA carboxylase (ACC) inhibitors, sodium-glucose cotransporter-2 (SGLT2) inhibitors, and fibroblast growth factor (FGF) analogues, are currently being tested in phase 2 or 3 clinical trials. Several late-stage clinical trials are also investigating the effects of agents that target the mechanisms involved in advanced stages of NAFLD, such as inflammation (C-C chemokine receptor CCR2/CCR5 antagonist cenicriviroc), apoptosis (caspase inhibitor emricasan, apoptosis signal-regulating kinase 1 ASK1 inhibitor selonsertib), and fibrosis (galectin-3 inhibitor belapectin). Given the multiple-hit pathogenesis of NAFLD, a multifactorial approach based on combination treatments simultaneously targeting several pathways (metabolic syndrome conditions, hepatic lipid accumulation, and NASH features) should be more effective than single drug therapy [211,212,213].

#### 3.2.4. Drugs Targeting Nuclear Receptors

Hepatic metabolic pathways, the alteration of which characterizes the first step of NAFLD, are mainly regulated at the transcriptional level. Therefore, transcription factors, and nuclear receptors in particular, may represent therapeutic targets in NAFLD. Within the nuclear receptor superfamily, PPARs, farnesoid X receptor (FXR), constitutive androstane receptor (CAR), pregnane X receptor (PXR), LXR, and thyroid hormone receptor-β (THR-β) are key regulators of the gut-liver-adipose tissue axis and control the expression of genes involved in lipid and glucose metabolism, bile acid homeostasis, and inflammation, which are all features of NAFLD/NASH [214,215,216]. Obeticholic acid is an FXR agonist that improves the histological features of NASH in patients without cirrhosis [217] and is currently being investigated in a phase 3 clinical trial [218]. An 18-month interim analysis of this ongoing study reported improved fibrosis in NASH patients treated with obeticholic acid compared to placebo [218]. A selective THR-β agonist, resmetirom, has demonstrated a highly significant reduction in hepatic fat and decreased hepatic inflammation in NASH patients following a 36-week treatment [219].

The three PPAR isotypes play distinct roles in lipid metabolism, energy homeostasis, and inflammation, which make them attractive targets in NAFLD, and they are discussed in more detail in the next section.

## 4. PPARs as Promising Targets for the Treatment of NAFLD

### 4.1. Overview of PPARs

PPARs are ligand-activated transcription factors belonging to the nuclear receptor family. Three isotypes of PPARs have been identified that are encoded by different genes: PPARα, PPARβ/δ, and PPARγ. Globally, the PPARs are activated by different ligands, have different tissue distribution, and distinct biological functions, but there is some overlap in these features (Table 1) and the three PPAR isotypes have a conserved protein structure and similar mechanisms of action (Figure 3). In addition, they all regulate energy homeostasis through lipid and glucose metabolism, and inflammation via modulation of largely specific target gene transcription.

#### 4.1.1. Structure, Tissue Expression, and Mode of Action

PPAR proteins contain four domains. The *N*-terminal A/B domain contains the ligand-independent transactivation function called activation function (AF)-1. The C domain is the DNA binding domain (DBD), which consists of two zinc-finger motifs that bind a specific DNA sequence called the peroxisome proliferator response element (PPRE), which is usually localized in gene promoters. The D domain is a flexible hinge region connecting the DBD and the ligand-binding domain (LBD). The C terminus domain contains the LBD and the ligand-dependent transactivation function AF-2, which includes the region for dimerization and interaction with regulatory proteins [220] (Figure 3A).

PPARα is highly expressed in oxidative tissues, such as the liver, skeletal muscle, brown adipose tissue (BAT), heart, and kidney. PPARβ/δ is most abundant in skeletal and cardiac muscles, adipose tissue, and skin, but also in inflammatory cells and liver cells, including hepatocytes, Kupffer cells, and HSCs. PPARγ is expressed predominantly in white and brown adipose tissue and macrophages [221] (Table 1).

PPARs have a large ligand-binding pocket, which contributes to their ability to bind various endogenous and synthetic ligands, as well as xenobiotics. The receptors are activated by endogenous ligands, including fatty acids and their derivatives, such as eicosanoids, which originate from dietary lipids, de novo lipogenesis, and adipose lipolysis [222,223,224,225,226] (Table 1). The development of several synthetic PPAR ligands, including molecules used in experimental research and pharmaceutical agents, has greatly contributed to the understanding of PPAR functions. Several studies reported that environmental pollutants also activate PPARs, supporting a role of PPARs in xenobiotic-induced toxicity in several organs [227,228,229].

All PPAR isotypes have a similar mechanism of action and function as heterodimers with the 9-cis retinoic acid receptor (RXR). In the absence of ligand, PPAR and its heterodimerization partner RXR are bound to corepressor complexes, leading to the repression of some target genes. Upon ligand binding to the LBD, a conformational change occurs, leading to corepressor dissociation and recruitment of coactivators. The activated PPAR/RXR heterodimers then bind to a DNA-specific sequence in the promoter of target genes (i.e., the PPRE) and stimulate transcription of the gene [226,230]. PPARs can also negatively regulate gene transcription via a PPRE-independent mechanism involving protein-protein interactions termed transrepression. In this process, PPARs bind other transcription factors, especially inflammatory transcription factors, inhibiting their binding to DNA and repressing their target gene transcription. Transrepression is the main mechanism involved in the anti-inflammatory effect of PPARs [231] (Figure 3B).

#### 4.1.2. PPARs in Glucose and Lipid Metabolism

Through modulation of gene transcription, the three PPAR isotypes play distinct roles in lipid and glucose metabolism, which are key processes in NAFLD pathogenesis.

PPAR expression and activity are regulated at several levels, including gene and protein expression, as well as ligand availability, post-translational modifications, and cofactor recruitment, and by different factors, such as hormones, cytokines, and growth factors [220,226,232]. Interestingly, hepatic expression of PPARs fluctuates in a circadian manner that is linked to the nutritional status [233]. For example, hepatic PPARα peaks in the early night, which corresponds to the end of the day-time fasting period in nocturnal rodents [234,235], whereas PPARβ/δ is active during the dark/feeding period [236]. Accordingly, PPARα is mainly active in the fasted state [235,237,238]. In response to fasting, hepatocyte PPARα controls the expression of several genes involved in whole-body fatty acid homeostasis, allowing the liver to use fatty acids and provide energy-rich fuel for other organs. PPARα facilitates fatty acid uptake by the liver and mitochondrial transport by controlling the transcription of genes encoding fatty acid transport proteins (fatty acid transport protein-1 [FATP1], CD36, fatty acid binding protein-1 [l-FABP]) and carnitine palmitoyltransferases (CPT1A, CPT2). PPARα is the central regulator of hepatic fatty acid catabolism, it regulates gene transcription of rate-limiting enzymes required for microsomal (cytochrome P450 family 4 subfamily A [CYP4A]), peroxisomal (acyl-CoA oxidase 1 [ACOX], enoyl-CoA hydratase and 3-hydroxyacyl CoA dehydrogenase [EHHADH]), and mitochondrial beta-oxidation (acyl-CoA dehydrogenase medium chain [ACADM], acyl-CoA dehydrogenase long chain [ACADL], acyl-CoA dehydrogenase very long chain [ACADVL]) [220,235,239]. In addition, hepatic PPARα regulates the expression of ketogenic enzymes, such as 3-hydroxy-3-methylglutaryl-CoA synthase 2 (HMGCS2), leading to the production of ketone bodies, which are a vital alternative source of energy in the absence of glucose for several organs, including the brain and heart [237,240]. Ketone bodies also act as cell signaling mediators and modulate inflammation [241]. Furthermore, PPARα is required for the hepatic expression of murine and human fibroblast growth factor 21 (FGF21) [242,243,244], an hepatokine with systemic metabolic effects and hepatoprotective properties [245]. Hepatocyte PPARα is also essential for fasting-induced angiopoietin-like protein 4 (Angptl4; inhibitor of lipoprotein lipase) expression, whereas expression of the genes encoding growth differentiation factor 15 (Gdf15) and Igfbp1 is increased in the absence of PPARα in hepatocytes [246,247]. During fasting, PPARα also increases the transcription of genes involved in autophagy, leading to lipophagy, a mechanism involved in hepatic lipid catabolism [248]. Interestingly, there is reciprocal regulation of PPARα and the autophagy-lysosomal signal [249]. Lysosomal inhibition leads to downregulation of PPARα and its target genes, decreasing peroxisomal lipid oxidation and biogenesis [250]. The class 3 PI3K, Vps15, which plays a central role in autophagy, has been shown to control PPARα activation for lipid degradation and mitochondrial biogenesis [251]. In hepatocytes, PPARα activation promotes lipoprotein TG hydrolysis by increasing the enzyme activity of lipoprotein lipase (LPL) through a direct increase in its transcription, and decreases the expression of genes encoding lipoproteins, such as apolipoprotein C3 and apolipoprotein A4, which act as inhibitors of LPL activity [220,225]. Consequently, activation of mouse and human PPARα reduces plasma TG levels, indirectly leading to increased plasma HDL-cholesterol levels and decreased plasma LDL-cholesterol levels [252]. A few studies have reported that, in the fed state, PPARα regulates hepatic lipogenesis, mainly indirectly through transcriptional upregulation of SREBP1c [253] and increased proteolytic cleavage into its active form [254]. PPARα also modulates glucose metabolism by regulating the expression of genes involved in hepatic glycerol metabolism, promoting gluconeogenesis [255], which could explain the marked hypoglycemia in fasted PPARα-deficient mice [238]. In addition to lipid and glucose metabolism, PPARα also regulates amino acid metabolism in the liver through regulation of the expression of enzymes involved in the transamination and deamination of amino acids and urea synthesis, which correlates with a modulation of the plasma urea concentration [256]. From the above information, regulation of the hepatic activity of PPARα is expected to impact liver physiology, especially lipid metabolism. One example of such regulation is that of the NAD+-dependent protein deacetylase sirtuin 1 (SIRT 1), which increases the activity of PPARα primarily through the activation of peroxisome proliferator-activated receptor-gamma coactivator 1α (PGC-1α). Deletion or overexpression of SIRT1 in hepatocytes decreases or increases the expression of PPARα target genes. Accordingly, hepatocyte-specific SIRT1-knockout mice fed a HFD develop liver steatosis, inflammation, and ER stress [257]. Lipid oxidation in the skeletal muscle [258] and white adipose tissue [259] is also controlled by PPARα. Overexpression of PPARα in the heart resulting in high PPARα-dependent fatty acid oxidation contributes to diabetic cardiomyopathy through a mechanism involving the cardiac lipoprotein lipase as a source of PPARα ligand [260]. Interestingly, PPARα-dependent regulation of fatty acid oxidation in extrahepatic tissues plays an important role during fasting and can compensate, at least in part, for the absence of PPARα in hepatocyte-specific *Ppara*-null mice [261]. A role for adipose PPARα in the β-adrenergic regulation of lipolysis has been suggested [262]. Overexpression of PPARα in adipose tissue is associated with improvement in HFD-induced alterations in glucose metabolism, mostly through modulation of BCAA metabolism [263]. The role of PPARα in brown adipose tissue thermogenesis and white adipose tissue browning remains unclear, as some studies have suggested that PPARα is required to maintain body temperature [235,264] and for adipocyte browning [265], whereas other studies indicate that PPARα is dispensable for cold-induced adipose browning [266] and brown adipocyte function in vivo [267]. Redundant roles of PPARα and PPARγ in brown adipose tissue may account for these discrepancies [268]. A recent study identified hepatocyte B-cell lymphoma 6 protein (BCL6) as a negative regulator of the PPARα-dependent transcription program during fasting. BCL6 interacts with a high number of the same genes as PPARα and represses lipid catabolism in the fed state [269]. Intriguingly, though PPARα is required for the adaptive response to fasting, it is dispensable during intermittent fasting, a condition that ameliorates hepatic steatosis [270]. Finally, PPARα has demonstrated interesting functions in hepatic sexual dimorphism. Its SUMOylation in the female liver causes repression of genes involved in steroid metabolism and immunity, which safeguards female mice against estrogen-induced intrahepatic cholestasis, the most common liver disease during pregnancy [271].

PPARβ/δ is well-studied in skeletal muscles [272], where its expression is induced by exercise training and promotes mitochondrial biogenesis and glucose uptake by increasing PGC-1α [273]. PPARβ/δ also increases PGC-1α expression, even after exercise cessation, by preventing its degradation [274]. In addition, PPARβ/δ is required to maintain oxidative fibers in muscles via the transcription of PGC-1α [275]. Transgenic mice overexpressing PPARβ/δ in adipose tissue are protected from HFD-induced obesity and exhibit decreased adipose lipid accumulation through thermogenic gene regulation [276]. In the liver, PPARβ/δ regulates both lipid and glucose metabolism [230]. Its expression is highly reduced by fasting and rapidly restored by refeeding [277]. PPARβ/δ activation improves insulin sensitivity in diabetic mice, mostly by regulating genes related to hepatic fatty acid synthesis and the pentose phosphate pathway [278]. Accordingly, liver PPARβ/δ overexpression through adenovirus improves glucose tolerance and insulin sensitivity in mice fed a HFD. PPARβ/δ regulates glucose utilization by increasing the transcription of genes involved in lipogenesis, glucose utilization, and glycogen synthesis through direct and indirect mechanisms [279]. Such indirect mechanisms include upregulation of the lipogenic transcription factor SREBP-1c and co-activator PGC-1β [279]. Intriguingly, hepatic PPARβ/δ overexpression leads to decreased liver damage, suggesting that it may protect from lipotoxicity by regulating MUFA synthesis [279]. In contrast, another study showed that PPARβ/δ regulates SREBP-1 activity via induction of insulin-induced gene-1 (Insig-1), which inhibits the proteolytic cleavage of SREBP-1 into its mature form and consequently leads to reduced lipogenesis [280]. PPARβ/δ also regulates the expression of genes involved in lipoprotein metabolism (*APOA4*, *VLDLR*) [281], which is consistent with the reduced plasma TG levels observed after PPARβ/δ ligand treatment [282,283]. PPARβ/δ deficiency induces an increase in VLDL receptor (VLDLR) levels and hepatic steatosis through the activating transcription factor 4 (ATF4) ER stress pathway [284]. Interestingly, *Pparβ/δ* deletion in CD11b+ Kupffer cells leads to hepatic lipid accumulation in early life, during the suckling period [230]. Recently, intestinal PPARβ/δ was shown to participate in reducing obesity, insulin resistance, and dyslipidemia in mice fed a HFD, but the underlying mechanism is unknown [285]. Notably, outside the scope of this review article, several aspects of PPARβ/δ function are relevant to cancer growth [286].

PPARγ is mainly active in the fed state and controls fat storage in adipose tissue. It transcriptionally regulates the expression of genes involved in adipogenesis and adipose differentiation, and in lipid metabolism, including fatty acid uptake (fatty acid binding protein 4 [FABP4], CD36) and TG lipolysis (LPL) in adipose tissues. Consequently, adipose-specific deficiency of PPARγ induces a dramatic loss of adipose tissue and severe insulin resistance, leading to hepatic fat accumulation [287,288]. PPARγ enhances insulin sensitivity not only by reducing adipose fatty acid influx into the liver, but also by inducing adipokines, such as adiponectin and leptin [192,193], as well as FGF1 [289]. A recent study indicated that adipose PPARγ also regulates the plasma levels of BCAA, which may participate in the insulin-sensitizing effects [290]. Another mechanism contributing to increased insulin sensitivity upon PPARγ activation is the induction of FGF21 in adipose tissue, which acts in an autocrine manner to reciprocally regulate PPARγ activity by suppressing its SUMOylation [291]. A more recent study indicated that PPARγ is required to maintain brown adipose tissue thermogenesis [267]. PPARγ expression in the liver is low under ordinary physiological conditions but increases during the development of steatosis in rodents. Hepatocyte-specific deletion of PPARγ in diabetic mice improves steatosis through decreased expression of lipogenic genes (fatty acid synthase (FASN), ACC, SCD1), but aggravates systemic insulin resistance, likely by decreasing insulin sensitivity in adipose tissue [292]. PPARγ also promotes hepatic lipid accumulation by regulating the expression of lipid-droplet-binding protein FSP27 [293,294]. The activator protein-1 (AP-1) complex is an important regulator of hepatic PPARγ signaling, and distinct AP-1 dimers differentially regulate human and mouse PPARγ transcription in the liver and, thus, hepatic lipid content [295]. In addition to lipid droplet formation, PPARγ is also involved in TG synthesis, which may prevent peripheral lipotoxicity by storing FFAs as TGs [296]. PPARγ activation can also promote hepatic steatosis induced by genetic insults through the upregulation of glycolytic enzymes (pyruvate kinase M2 (PKM2), hexokinase 2 (HK2)) [297].

Cross-talk between the different PPAR isotypes have been reported but are relatively little documented so far. The three PPAR isotypes contain a highly conserved DNA-binding domain and bind the same response element (PPRE) in the regulatory regions of target genes. Furthermore, they present overlapping expression patterns in several organs. Therefore, cross-talks between PPARs is likely. In fact, an interplay between PPARα and PPARγ has been reported in BAT. A set of genes involved in BAT function is activated by both a PPARα agonist (fenofibrate) and a PPARγ agonist (rosiglitazone) in mice, which suggests a functional redundancy, which may explain why some findings suggest that PPARα is dispensable for thermogenesis while others clearly indicate a role of PPARα in BAT function. As an example of redundancy, the gene coding for lysosomal protease cathepsin Z, a regulator of BAT thermogenic function, is a shared PPARα and PPARγ target gene [268]. Compensation between PPARs has also been observed. In PPARα-deficient mice fed a HFD, in which PPARγ is overexpressed in the liver, characteristic PPARα targets involved in fatty acid oxidation are up-regulated, indicating that PPARγ can compensate for PPARα in gene regulation [222]. Similarly, a compensatory role of PPARβ/δ in the repression of hepatic Cyp7b1 in female mice has been shown in the absence of PPARα [271]. Collectively, these studies reveal cross-talk and compensatory mechanisms between PPAR isotypes, which may be important to consider when testing PPAR agonists.

Overall, all three PPAR isotypes regulate lipid and glucose metabolism by regulating both overlapping and distinct genes in multiple organs [298] (Figure 4). PPARα is the master regulator of hepatic lipid catabolism in response to fasting. PPARγ promotes insulin sensitivity by controlling adipose lipid storage and adipocyte differentiation, whereas its role in the liver remains unclear. PPARβ/δ promotes hepatic glucose utilization and fatty acid synthesis, as well as fat catabolism in muscles.

#### 4.1.3. PPARs in Inflammation and HSC Activation

All PPARs play an important role in inflammation [299]. Evidence supports a role of PPARα in the control of hepatic inflammation [220]. One of the mechanisms by which PPARα exerts anti-inflammatory effects is through the down-regulation of acute phase genes and genes such as IL-1 receptor antagonist (IL-1Ra) and the nuclear factor kappa B subunit 1 (NF-κB) inhibitor IκB [220,300]. However, PPARα regulates inflammation mostly through a transrepression mechanism in which it binds to inflammatory transcription factors, such as NF-κB components (p65 and c-Jun), AP-1, and signal transducer and activator of transcription (STAT), thereby suppressing their transcriptional activity. An elegant study found that mice with a mutation in the DBD of PPARα, which limits its transcriptional activity to transrepression, are protected against liver inflammation through downregulation of pro-inflammatory genes and do not progress to liver fibrosis in dietary-induced NASH [301]. In addition, PPARα modulates the duration of inflammation by controlling the catabolism of its ligand leukotriene B4, a chemotactic agent involved in the inflammatory response [299,302]. Interestingly, hepatic PPARα contributes to the regulation of circulating monocytes during fasting through the modulation of bone marrow C-C motif chemokine ligand 2 (CCL2) production [303]. Few studies have examined the role of PPARα in Kupffer cells. A study of macrophage-specific PPARα-deficient mice compared to wild-type mice showed that Kupffer cell PPARα activation downregulates the expression of pro-inflammatory cytokines IL-15 and IL-18, which are mainly produced by M1 macrophages and Kupffer cells. This observation suggests that PPARα activation in these cells may prevent M1 polarization and mediate the anti-inflammatory effects of PPARα agonists [304]. Although anti-fibrogenic effects of PPARα activation have been reported in mouse models of liver fibrosis [301,305], the role of PPARα in HSCs is poorly defined. One study indicated that PPARα may inhibit TGFβ-stimulated HSC activation [306].

The role of PPARβ/δ in inflammation is less studied. PPARβ/δ is required for M2 macrophage activation in both adipose tissue and the liver. Bone marrow transfer experiments have shown that hematopoietic PPARβ/δ protects against HFD-induced insulin resistance, obesity, and fat accumulation in the liver. Moreover, PPARβ/δ appears to be necessary in Kupffer cells for oxidative phosphorylation, suggesting that these liver macrophages directly influence lipid homeostasis [307]. In addition, PPARβ/δ in CD11+ Kupffer cells has been suggested to prevent lipid accumulation in hepatocytes during the suckling stage in young mice [308]. PPARβ/δ is highly expressed in HSCs, but its role in these cells is not yet completely understood [230]. Studies in a mouse model of carbon tetrachloride (CCl4)-induced liver injury reported that PPARβ/δ activation stimulates HSC proliferation and promotes liver fibrosis [309], and contributes to HSC proliferation during acute and chronic hepatic inflammation in rats [310]. Another study in a mouse model of CCl4-induced liver fibrosis indicated that PPARβ/δ activation has anti-fibrotic effects [311]. The main difference between these studies is the use of different PPARβ/δ agonists.

PPARγ is expressed in macrophages, where it inhibits the expression of activated macrophage markers by reducing the activity of other transcription factors, including AP-1, STAT1, and NF-κB [312]. PPARγ activation induces monocyte differentiation towards M2 anti-inflammatory macrophages in vitro and in human blood [313]. In addition, PPARγ in macrophages is required for M2 macrophage activation and to protect mice against diet-induced obesity [314]. In line with these observations, PPARγ activation decreases HFD-induced M1 polarization through inhibition of the NF-κB pathway, reducing local inflammation and hepatic steatosis [315]. Interestingly, PPARγ promotes T regulatory cell accumulation in adipose tissue. Moreover, the PPARγ expressed by T regulatory cells is required for the insulin-sensitizing effect of PPARγ activation [316]. PPARγ expression and activation are reduced during HSC activation in vitro and in vivo [317]. In culture-activated HSCs, restoration of PPARγ levels using an adenoviral vector induces a phenotypic switch back to quiescence associated with inhibition of HSC activation markers [318]. Accordingly, ligand activation of PPARγ was shown to reduce HSC activation and proliferation, as well as collagen deposition, in a mouse model of CCl4-induced liver fibrosis [319]. The contribution of non-parenchymal cell PPARγ to the regulation of hepatic inflammation and fibrosis has been confirmed in the CCl4 model of liver injury [320].

To summarize, all PPARs play important roles in hepatic inflammation (Figure 4). PPARα negatively regulates pro-inflammatory genes, PPARβ/δ and PPARγ control macrophage M2 polarization. In addition, PPARγ has anti-fibrotic effects, but the roles of PPARα and PPARβ/δ in HSCs are not fully elucidated and require further study.

#### 4.1.4. PPARs in NAFLD

Human studies indicate a link between PPAR functions and NAFLD pathogenesis. In a cohort of obese patients with NAFLD, the hepatic expression of PPARβ/δ and PPARγ remained unchanged during NAFLD progression, but the expression of PPARα and its target genes negatively correlated with the histological severity of steatosis and NASH both at baseline and after 1 year of follow-up. In addition, decreased liver PPARα expression is associated with increased insulin resistance and decreased adiponectin levels [323]. More recently, reduced PPARβ/δ expression and activity were observed in patients with severe hepatic steatosis [284].

Preclinical evidence indicates a role of PPARs in mouse models of NAFLD. Hepatic expression of PPARα and its target genes is increased in mice undergoing chronic high-fat feeding. Interestingly, an increase in PPARγ expression has been observed in PPARα-deficient mice fed a HFD [222]. Whole-body PPARα-deficient mice develop obesity, which is more pronounced and associated with higher fat deposition in females [324]. PPARα deficiency in mice fed a HFD promotes hepatic steatosis and inflammatory gene expression [325,326]. In a mouse model of steatohepatitis induced by methionine and choline deficiency diet (MCD), PPARα-null mice develop more severe steatosis and steatohepatitis is associated with increased lipid peroxidation compared to control mice [327]. The systemic deletion of PPARα also leads to more severe steatosis in response to a trans fatty acid-rich diet [328]. Liver-specific PPARα-deficiency has revealed the importance of hepatocyte PPARα in protecting the animals from HFD-induced NAFLD, including steatosis and hepatic inflammation [235]. Interestingly, PPARα-null mice and hepatocyte-specific PPARα-deficient mice do not present with increased glucose intolerance when fed a HFD [329]. In addition, hepatocyte-specific deletion of PPARα induces spontaneous steatosis in aging mice and aggravates MCD-induced liver damage [235]. Interestingly, hepatocyte-specific depletion of G protein pathway suppressor 2 (Gps2), a co-repressor of PPARα, protects mice from HFD-induced steatosis and improves MCD-induced fibrosis through PPARα activation. In humans, liver Gps2 expression positively correlates with NASH and fibrosis [330]. In response to HFD, whole-body and hepatic deficiencies in PPARα differentially alter the lipid profiles, suggesting that extrahepatic PPARα is involved in lipid metabolism and the adaptive response to HFD [329]. PPARα in extrahepatic tissues also contributes to the protection of fasting-induced hepatosteatosis [261].

PPARβ/δ-deficient mice exhibit impaired thermogenesis and increased HFD-induced obesity [276]. PPARβ/δ deletion also leads to fat deposition in the liver and exacerbated hepatic steatosis induced by ER stress, which is accompanied by an increase in hepatic VLDLR levels [284]. In response to CCl4-induced liver toxicity, PPARβ/δ-deficient mice present with increased hepatotoxicity, which is associated with an increase in NF-κB signaling [331].

Systemic deletion of PPARγ induces embryonic lethality due to placental defect [332,333]. Recently, mice with whole-body PPARγ deletion except in the placenta were obtained. These mice are completely lipodystrophic, which is consistent with PPARγ being required in mature white and brown adipocytes for their survival [287], develop T2DM, and get a fatty liver [334]. As mentioned above, specific deletion of PPARγ in adipose tissue also leads to hepatic steatosis [288]. Increased hepatic expression of PPARγ is also observed in mice fed a HFD [335]. Intriguingly, hepatocyte-specific deletion of PPARγ protects mice against HFD-induced steatosis and glucose intolerance, but has no effect on insulin sensitivity, hepatic inflammation, or obesity [335,336]. In contrast, PPARγ deficiency in non-parenchymal liver cells (Kupffer cells and HSCs) aggravates acute and chronic CCl4-induced liver damage, increasing inflammatory and fibrogenic responses, whereas the deletion of PPARγ in hepatocytes does not have this effect [320]. Finally, a role of hepatic PPARγ in tumorigenesis has been shown in a mouse model of liver cancer [337].

Collectively, all PPAR isotypes regulate not only many aspects of glucose and lipid metabolism, but also contribute to anti-inflammatory responses, and potentially to HSC function (Figure 4). In addition, caloric restriction, which has beneficial effects in NAFLD patients, reduces the expression of PPARα and its target genes involved in lipid oxidation in the duodenum. Interestingly, this change in duodenum gene expression influences the microbiota composition [338]. Reciprocally, the gut microbiota appears to influence hepatic PPAR activity [339]. Moreover, some beneficial effects of gut microbiota on NAFLD were recently suggested to involve PPARs [340]. Overall, PPARs modulate the transcription of both overlapping and distinct downstream target genes involved in many NAFLD-related functions in multiple organs, including lipid and glucose metabolism and inflammation (Figure 4). Therefore, PPARs represent relevant targets for NAFLD.

### 4.2. Available PPAR Agonists

Several experimental and clinical studies have reported the use of PPAR agonists in the treatment of NAFLD [341,342], which we review below.

#### 4.2.1. PPARα Agonists

Fibrates are lipid-lowering agents used in clinical practice to treat hypertriglyceridemia and atherogenic dyslipidemia [343]. In rodent models of NAFLD, fibrates have demonstrated beneficial effects on hepatic steatosis, inflammation, and fibrosis. In MCD-induced mouse steatohepatitis, the PPARα agonist Wy14643 reduces hepatic TG levels and histological inflammation [327], as well as liver fibrosis in association with a decrease in HSC activation [305]. In this model, the beneficial effect of Wy14643 on MCD-induced liver damage is independent of its impact on fat accumulation in the liver, and due to the expression of genes involved in anti-inflammatory and anti-fibrogenic pathways [301]. PPARα activation by Wy14643 also decreases steatosis and inflammatory pathways in foz/foz mice, a genetic model of NASH, fed a HFD [344]. In the thioacetamide rat model of liver cirrhosis, Wy14643 and fenofibrate reverse histological liver fibrosis, in part by reducing the activity of the hepatic anti-oxidant enzyme catalase [345]. Fenofibrate also reduces CCl4-induced hepatic fibrosis in rats [346]. In a recent study, fenofibrate prevented liver damage induced by chronic intoxication of mice with 3,5-diethoxycarbonyl-1,4-dihydrocollidine (DDC), a model that induces key morphological features of NASH [347].

Conversely, fibrates only exhibit an effect on TG levels in humans. In obese patients with NAFLD, fenofibrate reduces plasma TGs by increasing VLDL-TG clearance from plasma, but does not change intrahepatic TG levels after 8 weeks of treatment [348]. Similarly, administration of fenofibrate for 48 weeks improves TG and glucose levels, but not liver histology in NAFLD patients [349]. Liver stiffness and biochemical markers of fibrosis (hyaluronic acid, TGF-β, and tumor necrosis factor-alpha (TNFα)) were decreased after 24 weeks of fenofibrate treatment, but no data on liver histology were given in this study [350]. In the Effects of Epanova Compared to Placebo and Compared to Fenofibrate on Liver Fat Content in Hypertriglyceridemic Overweight Subjects (EFFECT) I trial, 12 weeks of fenofibrate also reduced plasma TG levels, but increased liver fat content and liver volume in overweight or obese patients with NAFLD, suggesting a complex effect of fenofibrate on human hepatic lipid metabolism that requires further investigation [351]. Gemfibrozil has also demonstrated PPARα-dependent hypolipidemic actions [352] and attenuated hepatic lipid accumulation in vitro [353]. However, in NAFLD patients, gemfibrozil has only shown beneficial effects on plasma levels of liver enzymes [354,355].

#### 4.2.2. PPARβ/δ Agonists

Current PPARβ/δ agonists, including GW501516, GW0742, and MBX-8025 (Sedalpar), have mostly been tested in experimental models of NAFLD, and clinical studies are lacking. Though treatment of mice with GW501516 results in increased liver TG content after 4 weeks, long-term treatment (8 weeks) leads to reduced hepatic fat content. Interestingly, both PPARα and PPARβ/δ are required for the effect of GW501516 on hepatic lipid accumulation, as GW501516-dependent reduction in hepatic steatosis is abolished in PPARα-null mice. PPARβ/δ may modulate the levels of 1-palmitoyl-2-oleoyl-sn-glycero-3-phosphocholine (POPC), an endogenous activator of PPARα [356]. GW501516 treatment protects against HFD-induced obesity and insulin resistance, and reduces hepatic lipid accumulation by increasing muscle lipid oxidation [357]. GW501516 also increases the expression of hepatic VLDLR in mice fed a HFD [358]. Furthermore, GW501516 administration for 8 weeks decreases hepatic steatosis and insulin resistance in LDLR^−/−^ mice fed a HFD via the increased expression of genes involved in hepatic fatty acid oxidation and decreased expression of hepatic fatty acid synthesis genes [359]. However, GW501516 does not improve liver injury induced by CCl4 [311]. One human study reported that administration of GW501516 to healthy individuals for 2 weeks reduced liver fat content and serum TG levels [282].

In a diabetic rat model, GW0742 decreased hepatic TGs, glucose intolerance, epididymal fat weight, and inflammatory cytokines [360]. Another study indicated that GW0742 reduces hepatic TGs, glucose intolerance, and insulin resistance in mice fed a HFD. These effects were associated with several changes in hepatic gene expression, including an increase in PPARα and beta-oxidation gene expression and decreased expression of PPARγ and lipogenic genes, as well as genes involved in inflammation and ER stress [361]. GW0742 also reduces CCl4-induced hepatotoxicity, which is associated with modulation of NF-κB signaling [362].

The more recent PPARβ/δ agonist seladelpar reduces glucose intolerance and hepatic TGs in the foz/foz mouse model of NASH when fed an atherogenic diet. Seladelpar also decreased the NAS by 50% and reversed NASH in all mice. In addition, seladelpar improved liver histology, with decreased hepatic apoptosis and fibrosis and a reduction in the number of macrophages around hepatocytes (crown-like structures) [363].

#### 4.2.3. PPARγ Agonists

Thiazolidinediones (pioglitazone, rosiglitazone) are synthetic ligands of PPARγ that are clinically used as insulin sensitizers in the treatment of T2DM [193]. Though pioglitazone effectively improves hepatic steatosis in humans, preclinical data in rodents have been controversial, and the exact molecular mechanisms underlying the action of pioglitazone are not fully understood.

Several studies indicate that pioglitazone reduces HFD-induced steatosis in mice by increasing adiponectin production and the hepatic expression of genes involved in lipolysis, beta-oxidation, and autophagy [195,196]. In contrast, pioglitazone was shown to have no effect on liver histology in a rat dietary model of NASH (high fat, high cholesterol and cholate) [364]. A recent study showed that the effect of pioglitazone on NAFLD is influenced by CAR activity, as pioglitazone improves hepatic steatosis much better in CAR-deficient mice, suggesting an interaction between CAR and PPARγ [365]. Finally, another study indicates that pioglitazone promotes hepatic steatosis. In this study, the global expression profiles in the livers of mice fed a HFD and treated with pioglitazone reveal that pioglitazone upregulates the expression of genes involved in fatty acid uptake and de novo lipogenesis, and reduces the expression of inflammatory genes, leading to hepatic TG accumulation and improved insulin resistance [366].

As discussed above, several studies have reported that pioglitazone treatment is effective in NAFLD patients [94,180,181,183]. Pioglitazone improves the histological features of NAFLD, including steatosis and inflammation, whereas its effect on fibrosis is less clear. A recent meta-analysis reported that pioglitazone therapy is associated with an improvement in advanced fibrosis in NAFLD patients, even in non-diabetic patients. This meta-analysis also indicated that weight gain and limb edema is associated with pioglitazone treatment [367].

Another thiazolidinedione that has shown promising results in preclinical studies is rosiglitazone. In animal models, rosiglitazone protects against HFD-induced hepatic steatosis and reduces hepatic lipid content by increasing the expression of genes involved in beta-oxidation and decreasing the expression of lipogenic genes. These beneficial effects of rosiglitazone on lipid metabolism are accompanied by a decrease in hepatic M1 macrophages and modulation of the TLR4/NF-κB signaling pathways [368]. In the MCD model, rosiglitazone improves hepatic steatosis, inflammation, and fibrosis and reduces the expression of the HSC activator TGF-β [369]. In a model of liver cholestasis and fibrosis induced by bile duct ligation, rosiglitazone reduces fibrosis and hepatocyte apoptosis by inhibiting NF-κB-TNFα signaling in a PPARγ-dependent manner [370]. Interestingly, a recent study indicated that adipose PPARγ is dispensable for the whole-body insulin-sensitizing effect of rosiglitazone, suggesting the presence of PPARγ-independent targets of rosiglitazone in adipocytes, or that rosiglitazone activates PPARγ in other tissues [371]. In a small paired biopsy study, rosiglitazone treatment of NASH patients for 48 weeks results in improved hepatic steatosis, necroinflammation, and ballooning. In most patients, body weight increases during the treatment period, and the weight gain remains after a 6-month post-treatment follow-up [372]. The Fatty Liver Improvement With Rosiglitazone Therapy (FLIRT) trial assessed the effect of rosiglitazone in patients with biopsy-proven NASH. Treatment with rosiglitazone for 1 year increased adiponectin levels and reduced insulin resistance in most patients, and reduced hepatic steatosis in half of patients, but did not improve liver inflammation or fibrosis. The main adverse effect was weight gain in 40% of responders [373]. In a post hoc analysis of this cohort, patients treated with rosiglitazone presented with increased hepatic expression of PPARγ, which was associated with increased expression of several pro-inflammatory genes in the liver (monocyte chemoattractant protein-1 [MCP1], IL8, SOCS3), suggesting a potential long-term deleterious effect [374].

Although all PPAR agonists have had beneficial effects in preclinical models of NAFLD, their effectiveness in human pathology is limited. In NAFLD patients, PPARα activation only reduces plasma TG levels, whereas PPARγ agonists improve insulin sensitivity and steatosis, but do not seem to impact liver fibrosis. The efficacy of current PPARβ/δ agonists against NAFLD in humans is not known. Moreover, some PPAR agonists have adverse effects (weight gain and fluid retention following pioglitazone) or limited potency (fibrates) that limit their application.

Novel PPAR agonists, called selective PPAR modulators (SPPARMs), aim to maximize the beneficial effects and minimize the adverse effects of current agonists. Furthermore, given the multiple and distinct effects of PPARs in the liver and other organs, targeting two or three isotypes has emerged as a promising novel therapeutic strategy for treating NAFLD (Table 2).

### 4.3. Novel PPAR Agonists

#### 4.3.1. Pemafibrate

Pemafibrate (K-877) is a novel selective PPARα modulator that, compared to fenofibrate, exhibits high potency for human PPARα and enhanced PPARα selectivity and activity in vitro [375]. The crystal structure of the PPARα-pemafibrate complex showed that pemafibrate is highly flexible and can change its conformation following coactivator binding. In addition, several hydrophobic interactions between PPARα and pemafibrate may improve the binding affinity for PPARα [376]. The hepatic transcriptome of primary human hepatocytes and mice treated with pemafibrate indicates a PPARα-dependent increase in the expression of genes involved in lipid catabolism and ketogenesis. Interestingly, VLDLR and FGF21 are also induced by pemafibrate in humans and mice, and at a higher level than by fenofibrate [377,378]. Pemafibrate decreases plasma TG and total cholesterol levels in the LDLR^−/−^ mouse model of atherosclerosis, which is associated with increased expression of PPARα and its target genes in both the liver and intestine [379]. In Western diet-fed ApoE2KI mice, pemafibrate also improves lipoprotein metabolism, resulting in a greater reduction in TG and increase in HDL-cholesterol levels compared to fenofibrate. Pemafibrate also decreases atherosclerotic lesions, lesion macrophage infiltration, and inflammatory markers [380]. In mice fed a HFD, pemafibrate reduces postprandial accumulation of TGs at the same level as fenofibrate, but at lower doses [381]. In addition, pemafibrate protects against HFD-induced obesity, glucose intolerance, and insulin resistance, and decreases the cell size in white adipose tissue and brown adipose tissue, but has no effect on hepatic TG accumulation. Pemafibrate-activated PPARα in the liver increases hepatic and plasma levels of FGF21, whereas in inguinal adipose tissue, pemafibrate increases the expression of genes involved in fatty acid oxidation and thermogenesis, and the mitochondrial marker elongation of very long chain fatty acids protein 3 (Elovl3) in brown adipose tissue [382]. One study examined the effects of pemafibrate in a mouse model of NASH induced by an amylin diet that exhibits the different stages of NASH, including steatosis, inflammation, hepatocyte ballooning, and fibrosis. Pemafibrate reduces hepatic TG levels, inflammation, and fibrosis, and increases expression of PPARα and its target genes involved in beta-oxidation. In addition, pemafibrate increases the expression of lipogenic genes. However, in contrast to previous reports, fenofibrate tested in parallel with pemafibrate was also effective in reducing fibrosis [383]. Recently, the therapeutic potential of pemafibrate was tested in a mouse model of a diabetes-based NASH-HCC model. In this model, combined chemical (one low dose of streptozotocin just after birth) and dietary (continuous HFD feeding) interventions leads to diabetes in 1 week and sequential liver damage from steatosis, NASH, and HCC but did not induce obesity and insulin resistance. Pemafibrate reduces macrophage recruitment and inflammation in the liver, potentially through the downregulation of endothelial adhesion molecules. Intriguingly, hepatic TG accumulation is not improved with pemafibrate in this model [384].

In clinical studies, pemafibrate has demonstrated safety and efficacy in patients with atherogenic dyslipidemia [385] and appears to be superior to fenofibrate to reduce plasma TG levels [386,387,388]. In a phase 3 clinical trial, treatment of Japanese T2DM patients with pemafibrate for 24 weeks reduced fasting serum TG levels by 45%. Interestingly, in this cohort, pemafibrate increased plasma FGF21 [389]. The ongoing Pemafibrate to Reduce Cardiovascular Outcomes by Reducing Triglycerides in Patients with Diabetes (PROMINENT) study is a placebo-controlled trial testing the effect of pemafibrate on cardiovascular events in T2DM patients with elevated TG and low HDL-cholesterol levels [390].

Due to the multiple and distinct effects of PPARs, dual or pan-PPAR agonists represent attractive approaches for targeting the multiple biological processes involved in the pathogenesis of NAFLD. Below, we review the most promising of them.

#### 4.3.2. PPARα and β/δ Dual Agonist Elafibranor

The PPARα and PPARβ/δ dual agonist elafibranor (GFT-505) has preferential activity on human PPARα in vitro and additional but lower activity on human PPARβ/δ [391]. Several experimental studies indicate that elafibranor has beneficial effects on NAFLD/NASH and fibrosis in rodent models. Efficiency was first demonstrated in Western diet-fed human apolipoprotein E2 transgenic mice, in which elafibranor improves lipid profiles and reduces hepatic expression of pro-inflammatory and pro-fibrotic genes. Histological examination has demonstrated that elafibranor decreases steatosis, inflammation, and fibrosis. Similar results have been reported in ob/ob mice fed MCD. Using PPARα-deficient mice, this study demonstrates the importance of PPARα in the effects of elafibranor, but also reveals PPARα-independent mechanisms [392]. In other mouse models of diet-induced NASH, elafibranor induces weight loss and improves steatosis, as well as inflammation and fibrosis. Hepatic transcriptome analysis has revealed that elafibranor modulates the expression of genes involved in lipid metabolism, inflammation, fibrogenesis, HSC activation, and apoptosis [393,394]. Elafibranor has also shown efficiency in a rapid diet-induced NASH model with additional cyclodextrin in drinking water, which induces NASH in 3 weeks without obesity [395]. Elafibranor also prevents and reverses CCl4-induced liver fibrosis and inflammation in rats [392]. Interestingly, in alcoholic steatohepatitis, elafibranor reduces adipose tissue autophagy dysfunction, leading to hepatoprotective and anti-inflammatory effects in several organs, including the liver, intestines, and adipose tissue [396].

Few clinical studies have reported the impact of elafibranor in humans. Eight-week treatment with elafibranor reduces fasting plasma TG levels and improves both hepatic and whole-body insulin sensitivity in obese insulin-resistant patients. Elafibranor also improves liver enzyme levels, suggesting beneficial effects on liver functions [397]. A phase 2 study examined the efficacy of elafibranor treatment for 1 year in NASH patients. According to the updated definition of resolution for NASH, elafibranor resolves NASH without fibrosis worsening, but only in patients with severe disease (NAS > 4). Elafibranor is not efficient in patients with mild disease (NAS < 4) and fails to demonstrate a beneficial effect on fibrosis [398]. In these human studies, elafibranor had a safety profile with no specific adverse effects.

The combination of its insulin-sensitizing and hepatoprotective effects makes elafibranor a good candidate for treating NAFLD. However, no beneficial effects on fibrosis have been demonstrated. It is currently being tested in a phase 3 clinical trial in NASH patients with fibrosis (NAS > 4).

#### 4.3.3. PPARα and γ Dual Agonist Saroglitazar

Glitazars are dual PPARα/γ agonists developed to combine the beneficial effects of PPARα on plasma TGs and lipoproteins and PPARγ on insulin resistance. Most of these agonists have been discontinued due to adverse effects, but saroglitazar (Lipaglyn) was clinically approved in India in 2013 to treat diabetic dyslipidemia [399]. In vitro, saroglitazar has predominant activity on PPARα and moderate activity on PPARγ, reducing the adverse effects associated with PPARγ activation by pioglitazone [400].

In a diet-induced mouse model of NASH (choline-deficient, l-amino acid-defined, HFD), saroglitazar leads to a greater reduction in NAS than the PPARα agonist fenofibrate and PPARγ agonist pioglitazone. Histological examination of the liver tissue demonstrated a strong reduction in steatosis and decreased hepatocyte ballooning and inflammation, but only a trend in reduced fibrosis. Saroglitazar reduces hepatic expression of pro-inflammatory and pro-fibrotic genes. In vitro, saroglitazar decreases lipid-mediated oxidative stress and HSC activation. In addition, saroglitazar reduces CCl4-induced liver fibrosis in rats [401] and regulates adipose tissue homeostasis in mice [402]. In HFD-fed mice, saroglitazar improves serum TG levels and insulin resistance and reduces body weight and white adipose tissue mass. Histological examination of the adipose tissue has shown that saroglitazar reduces adipocyte hypertrophy by increasing the expression of thermogenic genes. In addition, saroglitazar treatment increases M2 macrophages and decreases M1 macrophages in adipose tissue, indicating that saroglitazar promotes an anti-inflammatory environment in adipose tissue [402]. In a rapid rat model of NASH induced by high-fat emulsion and small doses of LPS, saroglitazar improved adipocyte dysfunction through increased plasma adiponectin. In the liver, saroglitazar induced a decrease in TLR4 signaling upon LPS administration, with reduced NF-κB, TLR4, and TGFβ, which suggests a role of saroglitazar in response to gut endotoxins [403]. Saroglitazar also reduces thioacetamide-induced liver fibrosis in rats and decreases leptin, TGF-β, and platelet-derived growth factor (PDGF-BB) in the liver [404].

Several clinical studies have indicated that saroglitazar treatment in patients with diabetic dyslipidemia results in improved lipid and glucose parameters, including a reduction in plasma TG levels and fasting plasma glucose [405,406], and improves whole body insulin sensitivity in these patients [407]. In a review summarizing 18 studies on the effect of saroglitazar in patients with diabetic dyslipidemia, saroglitazar treatment was associated with a reduction in ALT levels and fatty liver in NAFLD patients with diabetic dyslipidemia [408]. A phase 2 study is evaluating the safety and efficacy of saroglitazar in patients with NASH. The primary endpoint is to assess the changes in NAS with no worsening of fibrosis from baseline to week 24 of treatment. The 16-week efficacy of saroglitazar in reducing serum ALT in NAFLD patients is also being tested in a phase 2 trial. Furthermore, two phase 3 clinical trials are currently evaluating saroglitazar in NAFLD with an amelioration of the fibrosis score as the primary outcome. The first study is investigating the efficacy of saroglitazar compared to pioglitazone in NAFLD patients over a period of 24 weeks. The second study is comparing the effect of combined saroglitazar and vitamin E treatment vs. vitamin E alone vs. saroglitazar alone (NCT04193982). Based on all observations thus far, saroglitazar shows promise as a potential NASH drug.

#### 4.3.4. Pan-PPAR Agonist Lanifibranor

Lanifibranor (IVA337) is a moderately potent and well-balanced modulator of the three PPAR isotypes and has a good safety profile. Compared to glitazones, lanifibranor has demonstrated differences in co-regulator recruitment [409]. Treatment of db/db mice with lanifibranor induces a dose-dependent decrease in circulating glucose and TG levels [409]. Lanifibranor has also been tested in mouse models of NASH [410,411]. In the MCD model, lanifibranor reduces steatosis and hepatic TG levels, as well as inflammation. In foz/foz mice fed a HFD, lanifibranor attenuates steatosis, inflammation, and hepatocyte ballooning. Hepatic gene expression analysis has shown increased expression of genes involved in fatty acid beta-oxidation and decreased expression of pro-inflammatory and pro-fibrotic genes. In addition, lanifibranor treatment improves metabolic parameters, such as glucose intolerance, and increases plasma adiponectin levels [410]. Beneficial effects of lanifibranor on NASH histology, including reduced fibrosis, were confirmed recently in a preclinical model of NASH and fibrosis (choline-deficient amino acid-defined HFD mouse model). Interestingly, decreased macrophage infiltration in the liver has been observed upon lanifibranor treatment, suggesting that Kupffer cells may be important targets of lanifibranor to improve NAFLD. Similar results of NASH histology were obtained in the Western diet model, together with a reduction in plasma TG levels [411]. Lanifibranor is also effective in reducing collagen deposition and increasing plasma adiponectin in mice with CCl4-induced liver fibrosis [409,410]. In vitro results have demonstrated that lanifibranor inhibits the proliferation and activation of HSCs, as well as the activation of hepatic macrophages [410,411]. The anti-inflammatory and anti-fibrotic effects of lanifibranor were also demonstrated in preclinical mouse models of skin and pulmonary fibrosis [412,413].

Lanifibranor is currently undergoing phase 2 clinical trials in NAFLD. The first study is evaluating the efficacy and the safety of two doses of lanifibranor for 24 weeks vs. placebo in adult NASH patients with liver steatosis and moderate to severe necroinflammation without cirrhosis. The second study is designed to study lanifibranor in patients with T2DM and NAFLD.

The therapeutic potential of the novel PPAR agonists discussed above appears to be well-established in experimental models. However, none of the current preclinical models of NASH reproduce all features of human NASH [414]. In addition, differences exist between humans and mice regarding the PPARs. For example, hepatic PPARα expression is higher in rodents than in humans, which may explain why PPARα activation has stronger beneficial effects in rodent NAFLD. In this respect, “humanized” preclinical strategies, for example using transgenic mice expressing human PPARs or mice with a humanized liver may represent a relevant strategy for the evaluation of PPAR agonists [342]. In addition, a better understanding of the molecular mechanisms, especially the transcriptional coregulator network, underlying PPAR-dependent transcription in different species, tissues and diseases may be needed for designing more-specific and more-potent PPAR ligands [415]. Finally, PPAR functions are also regulated at the level of posttranslational modifications that influence protein stability and localization, ligand binding, and co-factor interaction. Understanding the role of these posttranslational modifications and their association with diseases might help in the development of novel molecules that specifically inhibit or promote such modifications [232].

In a relatively near future, the results of phase 2 and phase 3 clinical trials will determine the therapeutic potential of these novel compounds in NAFLD. Given the role of PPARs in multiple pathways involved in NAFLD and the beneficial effects of each single isotype agonist, we consider combined activation of several PPARs as a promising approach for NAFLD treatment because of potential optimization of the benefits and reduction of the side effects (Figure 5).

## 5. Concluding Remarks

The prevalence of NAFLD is dramatically increasing in developed countries, but no approved therapy is available. Most of the current pharmacological strategies target comorbidities, such as the manifestations of metabolic syndrome. Vitamin E and pioglitazone have beneficial effects on steatosis and inflammation, but can induce adverse effects in some patients. In addition, none of the currently used medications improve fibrosis, which is the strongest indicator of mortality in NAFLD. As highlighted in this review, the pathogenesis of NAFLD is multifactorial, which represents both a challenge and an opportunity for developing intervention strategies. As regulators of gene expression, the three PPARs impact, in some way, all currently known functions associated with NAFLD pathogenesis. The PPARs have emerged as crucial regulators of the whole organism and cellular metabolic functions. As links between lipid signaling and inflammation, they also fine-tune the crosstalk between metabolic processes and the innate immune system. All of these attributes make them relevant targets for treating NAFLD. Different approaches may be successful. One approach would be to selectively modify the pharmacological characteristics of agonists, as has been done with the PPARα selective modulator pemafibrate, to ameliorate the profile of beneficial effects with respect to issues associated with fibrate treatment. Developing molecules simultaneously targeting two or all three PPAR isotypes is another promising approach for NAFLD treatment that allows targeting of the multifaceted roles of PPARs. The dual agonists—elafibranor and saroglitazar—and the pan agonist lanifibranor have demonstrated many beneficial effects on liver histology with minimal adverse effects. Some of these novel agonists are currently in phase 3 clinical trials and appear promising for NASH treatment. As discussed herein, PPARs not only impact liver, but also other organs. In particular, they can have both positive and negative effects on heart physiology, pathology and injury [416,417]. Therefore, it is of importance that the new potential NASH drugs are evaluated for potential beneficial as well as deleterious effects on cardiac functions. In a post hoc analysis, elafibranor resolved NASH without fibrosis worsening and did not cause cardiac events [398]. Saroglitazar showed a potential to lower the cardiovascular risk in T2DM patients [418]. Pemafibrate is currently being tested for its effect on reducing cardiovascular events in diabetic patients with high TG levels in the PROMINENT study (NCT03071692) [390].

In parallel with the study of these novel promising agonists, it will be important to increase knowledge of the liver-specific functions of the PPAR isotypes, particularly in hepatocytes, Kupffer cells, and HSCs, and deepen our understanding of their roles in inflammation and fibrosis. Drugs that combine PPAR activation and other PPAR-independent pathways, which converge in ameliorating the manifestations of NAFLD, are also worth exploring. Interestingly, telmisartan, an angiotensin receptor blocker, is also a PPARα/γ dual agonist and worth exploring in the treatment of NAFLD. Inhibition of angiotensin converting enzyme (ACE) and angiotensin II type 1 receptor (AT1) improves liver fibrosis by keeping HSCs in a quiescent state through suppression of TGF-β [419]. These effects, combined with those known of PPARα/γ activation, deserve further investigation. Given the importance of circadian clock proteins in coordinating energy metabolism, the clock regulation of drug targets should also be taken into account in the development of pharmaceuticals for the treatment of NAFLD [420]. Despite recent remarkable and fast progress in the field, there are still many challenges imposed by the complex physiopathology underlying the development and progression of NAFLD, not least of all its heterogeneity, which is not fully understood. For example, why some patients will progress to advanced stages and others will not is not clear, and NAFLD in lean patients is also not completely understood.

## Figures and Tables

**Figure 1 cells-09-01638-f001:**
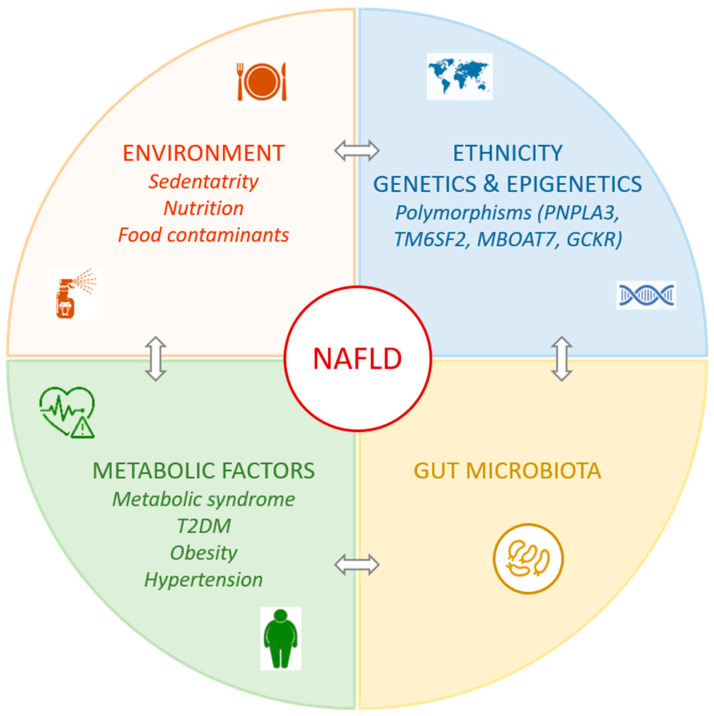
Non-alcoholic fatty liver disease (NAFLD) determinants. Multiple factors contribute to the development of NAFLD and its progression. Obesity and T2DM are closely associated with NAFLD, and both drive the increasing prevalence of NAFLD. The genetic background also strongly influences disease development. In addition, the progression of NAFLD depends on complex interactions between genetic and environmental factors, especially dietary factors. More recently, the gut microbiota has emerged as an important determinant of NAFLD pathogenesis. Abbreviations: NAFLD, non-alcoholic fatty liver disease; T2DM, type 2 diabetes mellitus; PNPLA3, patatin-like phospholipase domain containing protein 3; TM6SF2, transmembrane 6 superfamily 2; MBOAT7, membrane bound O-acyltransferase domain-containing 7; GCKR, glucokinase regulator.

**Figure 2 cells-09-01638-f002:**
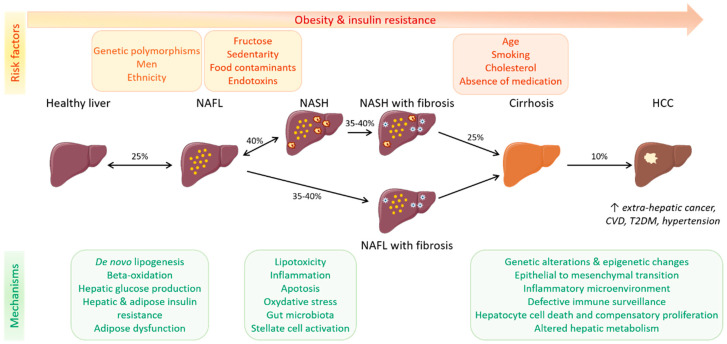
NAFLD progression. NAFLD is a progressive disease characterized by fat accumulation in hepatocytes, ranging from hepatic steatosis (NAFL) to non-alcoholic steatohepatitis (NASH), with additional inflammation with or without fibrosis. The latter is the strongest histological predictor of disease-related mortality. Though steatosis has previously been considered to be benign, some NAFL patients progress to NASH with or without fibrosis, whereas others develop fibrosis without having NASH. The pathogenesis of NAFLD is complex and involves several different pathways in multiple organs, including metabolic and inflammatory pathways. Abbreviations: NAFL, non-alcoholic fatty liver; NASH, non-alcoholic steatohepatitis; HCC, hepatocellular carcinoma; T2DM, type 2 diabetes mellitus; CVD, cardiovascular disease.

**Figure 3 cells-09-01638-f003:**
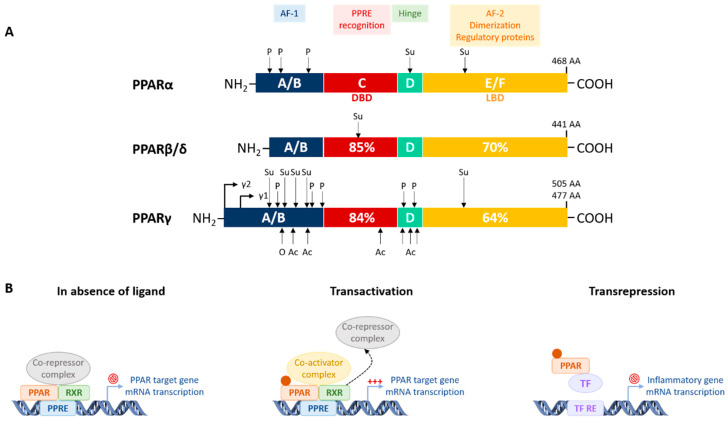
Protein structure and mechanisms of transcriptional regulation of PPARs. (**A**) Functional domains and posttranslational modifications of human PPARs. PPARs contain four distinct domains: a *N*-terminal A/B domain (ligand-independent AF-1), a C-domain (DNA-binding domain), a D- domain (hinge domain), and a *C*-terminal E/F domain (ligand-binding domain). Main functions of the 4 domains are listed. The number inside each domain corresponds to the percentage of amino acid sequence identity of human PPARβ/δ and γ relative to PPARα. The number of amino acids indicated at the COOH-terminus are for the human receptors. The locations of posttranslational modification sites are indicated by arrows. The 2 splice variants of PPARγ are indicated by γ1 and γ2. (**B**) PPAR mechanism of action. In the absence of ligand, PPAR-RXR heterodimers are bound to corepressor complexes and prevent gene transcription. Binding of an endogenous ligand or a synthetic agonist to the PPAR LBD triggers a conformational change, leading to corepressor complex dissociation and recruitment of coactivator complex. The activated PPAR/RXR heterodimer then binds to a specific DNA sequence in the promotor region of target genes (PPRE) and stimulates target gene transcription (transactivation). Through the binding to inflammatory transcription factors such as NF-κB and AP-1 (identified by TF), PPARs inhibit their binding to DNA and negatively regulate expression of proinflammatory genes (transrepression). Abbreviations: AF-1, activation function-1; AF-2, activation function-2; PPRE, peroxisome proliferator response element; DBD, DNA binding domain; LBD, ligand binding domain; AA, amino acid; P, phosphorylation; Su, SUMOylation; Ac, acetylation; O, *O*-GlcNacylation; RXR, 9-*cis* retinoic acid receptor; TF, transcription factor; TF RE, transcription factor response element; NH_2_, protein N terminus; COOH, protein carboxyl terminus.

**Figure 4 cells-09-01638-f004:**
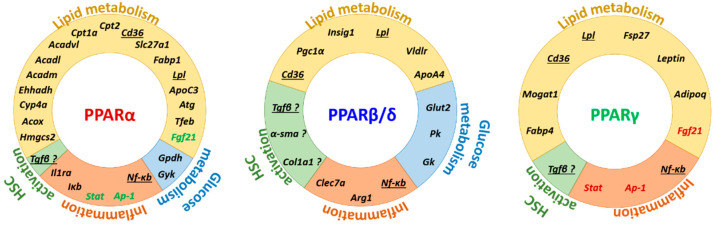
PPAR target genes and their implications in major functions associated with NAFLD pathogenesis. Key target genes of PPARα (**A**), PPARβ/δ (**B**), and PPARγ (**C**) and their association with four main biological processes driving NAFLD development and progression, i.e., lipid and glucose metabolism, inflammation, and hepatic stellate cell (HSC) activation. Genes in red, blue, and green are also regulated by PPARα, PPARβ/δ, and PPARγ, respectively. Genes whose expression is regulated by all three PPARs are underlined. A question mark (?) after a gene name indicates that PPAR may potentially regulate it. (**A**) PPARα promotes the expression of genes involved in fatty acid catabolism (Hmgcs2, Acox, Cyp4a, Ehhadh, Acad, Cpt1a, Cpt2, Cd36, Slc27a1, Fabp1) [235,237,321,322], autophagy (Atg, Tfeb) [248,249], and glycerol metabolism (Gyk, Gpdh) [255], and regulates lipoprotein metabolism (Lpl, ApoC3) [220] and hepatic Fgf21 expression [242,243]. It also downregulates inflammatory genes and transcription factors (Nf-κb, Ap-1, Stat, Iκb, Il1ra) [220,281,300], and may downregulate Tgfβ expression [306]. (**B**) PPARβ/δ increases PGC-1α in muscles [273,275], Insig1 [280] and Cd36 [279] in the liver, and regulates the expression of genes involved in lipoprotein metabolism (ApoA4, Vldlr) [281,284] and glucose utilization [279]. PPARβ/δ regulates the expression of genes induced during alternative macrophage activation (Arg1, Clec7a) [307] and may also influence HSC activation (Tgfβ, α-Sma, Col1a1) [309,310]. (**C**) PPARγ controls the expression of genes involved in adipogenesis (Fabp4, Cd36, Lpl, Mogat1) [296,314] and genes encoding adipokines (Adipoq, leptin, Fgf21) [192,193,289,291]. It also promotes Fsp27 expression in the liver during steatosis [293]. PPARγ downregulates inflammatory transcription factors (Nf-κb, Ap-1, Stat) [312,315] and may also reduce expression of Tgfβ [318,319]. Abbreviations: Hmgcs2, 3-hydroxy-3-methylglutaryl-CoA synthase 2; Acox, peroxisomal acyl-coenzyme A oxidase 1; Cyp4a, cytochrome P450 family 4 subfamily A; Ehhadh, enoyl-CoA hydratase and 3-hydroxyacyl CoA dehydrogenase; Acadm, acyl-CoA dehydrogenase medium chain; Acadl, acyl-CoA dehydrogenase long chain; Acadvl, acyl-CoA dehydrogenase very long chain; Cpt1a, carnitine palmitoyltransferase 1a; Cpt2, carnitine palmitoyltransferase 2; Slc27a1, solute carrier family 27 member 1; Fabp1, fatty acid binding protein 1; Fabp4, fatty acid binding protein 4; Lpl, lipoprotein lipase; ApoC3, apolipoprotein C3; ApoA4, apolipoprotein A4; Atg, autophagy-related genes; Tfeb, transcription factor EB; Fgf21, fibroblast growth factor 21; Pgc1α, peroxisome proliferator-activated receptor gamma coactivator 1-alpha; Fasn, fatty acid synthase; Acc, acetyl-CoA carboxylase; Scd1, stearoyl-CoA desaturase; Insig1, insulin-induced gene 1; Vldlr, very-low density lipoprotein receptor; Mogat1, monoacylglycerol O-acyltransferase 1; Fsp27, fat specific protein 27; Adipoq, adiponectin; Gpdh, glyceraldehyde-3-phosphate dehydrogenase; Gyk, glycerol kinase; Glut2, glucose transporter type 2; Pk, pyruvate kinase; Gk, glucokinase; Nf-κb, nuclear factor kappa B subunit 1; Ap-1, activator protein; Stat, signal transducer and activator of transcription; Iκb, Nf-κb inhibitor; Il1ra, IL-1 receptor antagonist; Arg1, arginase 1; Clec7a, *C*-type lectin domain containing 7A; Tgfβ, transforming growth factor beta; Col1a1, collagen type I alpha 1 chain; α-Sma, alpha-smooth muscle actin.

**Figure 5 cells-09-01638-f005:**
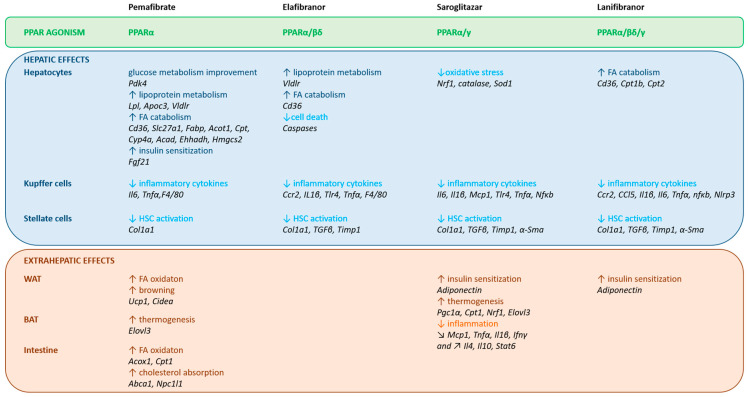
Novel PPAR agonists with potential for NAFLD treatment. The main genes regulated by the PPARα agonist pemafibrate, the dual PPARα and β/δ agonist elafibranor, the dual PPARα and γ agonist saroglitazar, and the pan-PPAR agonist lanifibranor in preclinical models of NAFLD. Through the modulation of gene expression, these novel compounds regulate several hepatic and extrahepatic pathways, including lipid and glucose metabolism, inflammation, and hepatic stellate cell activation, which are all key processes involved in NAFLD. Abbreviations: BAT, brown adipose tissue; WAT, white adipose tissue; Pdk4, pyruvate dehydrogenase kinase 4; Lpl, lipoprotein lipase; ApoC3, apolipoprotein C3; Vldlr, very-low density lipoprotein receptor; Cpt, carnitine palmitoyltransferase; Slc27a1, solute carrier family 27 member 1; Fabp, fatty acid binding protein; Cyp4a, cytochrome P450 family 4 subfamily A; Ehhadh, enoyl-CoA hydratase and 3-hydroxyacyl CoA dehydrogenase; Acad, acyl-CoA dehydrogenase; Hmgcs2, 3-hydroxy-3-methylglutaryl-CoA synthase 2; Acot1, acyl-CoA thioesterase 1; Fgf21, fibroblast growth factor 21; Nrf1, nuclear respiratory factor 1; Sod1, superoxide dismutase 1; Il1β, interleukin-1 beta; Il6, interleukin-6; Tnfα, tumor necrosis factor-alpha; Ccr2, C-C motif chemokine receptor 2; Tlr4, Toll-like receptor 4; Mcp1, monocyte chemoattractant protein 1; Ccl5, C-C motif chemokine ligand 5; Nlrp3, NLR family pyrin domain containing 3; Nfκb, nuclear factor kappa B subunit 1; Tgfβ, transforming growth factor beta; Col1a1, collagen type I alpha 1 chain; Timp1, metalloproteinase inhibitor 1; α-Sma, alpha-smooth muscle actin; Ucp1, uncoupling protein 1; Cidea, cell death inducing DFFA like effector A; Elovl3, elongation of very long chain fatty acids protein 3; Acox, peroxisomal acyl-coenzyme A oxidase 1; Abca1, ATP binding cassette subfamily A member 1; Npc1l1, Niemann-Pick C1-like protein 1; Pgc1α, peroxisome proliferator-activated receptor gamma coactivator 1-alpha; Ifnγ, interferon gamma; Il4, interleukin-4; Il10, interleukin-10; Stat6, signal transducer and activator of transcription 6.

**Table 1 cells-09-01638-t001:** Expression, ligands, and functions of Peroxisome proliferator-activated receptors (PPARs) related to NAFLD and therapeutic potential.

Isotypes	PPARα	PPARβ/δ	PPARγ
**Main tissue expression**	Liver	Skeletal & cardiac muscles	WAT
	Skeletal muscles		BAT
	Heart	Liver	Macrophages
	Kidney	WAT	
	BAT	BAT	
	Intestine	Macrophages	
**Main natural ligands**	FA	FA	FA
	Eicosanoids	VLDL components	Arachidonic acid metabolites
	Phospholipids		
**Main synthetic single agonists**	Fenofibrate	GW501516	Pioglitazone
	Wy14643	GW0742	Rosiglitazone
	Gemfibrazil	Seladelpar	
	Pemafibrate		
**Biological functions related to NAFLD**	TG hydrolysis	Muscle FA storage	Adipogenesis
	FA catabolism	FA catabolism	Adipose FA storage
	Ketogenesis	Lipoprotein metabolism	Adipokine secretion
	FGF21 production	Glucose utilization	Anti-inflammatory
	Glycerol metabolism	Anti-inflammatory	Anti-fibrotic
	Anti-Inflammatory		
**Potential therapeutic target for**	Hypertriglyceridemia	Atherogenic dyslipidemia	Insulin resistance
	Atherogenic dyslipidemia	Insulin resistance	Obesity
	NAFLD	Obesity	T2DM
		T2DM	NAFLD
		NAFLD	

Abbreviations: BAT, brown adipose tissue; WAT, white adipose tissue; FA, fatty acid; VLDL, very low density lipoprotein; TG, triglyceride; FGF21, fibroblast growth factor 21; T2DM, type 2 diabetes mellitus; NAFLD, non-alcoholic fatty liver disease.

**Table 2 cells-09-01638-t002:** PPAR agonists currently in late-stage clinical trials (phase 2 and phase 3). Overview of new PPAR agonists: trivial name, chemical structure, and short description.

Compounds	Chemical Structure	Description
**Pemafibrate**	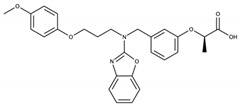	Selective and potent PPARα modulator
		Clinical application for dyslipidemia
		Safety profile in clinical studies
		Currently in phase 3: effect on CV events in T2DM
**Elafibranor**	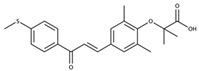	High activity on PPARα, moderate activity on PPARβ/δ
		Safety profile in clinical studies
		Currently in phase 3: effect on fibrosis in NASH
**Saroglitazar**	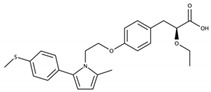	High activity on PPARα, moderate activity on PPARγ
		Clinical application for dyslipidemia
		Safety profile in clinical studies
		Currently in phase 2: effect on NAFLD/NASH, phase 3: effect on fibrosis compared to pioglitazone and vitamin E
**Lanifibranor**	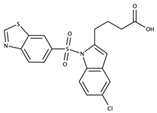	Moderate and balanced activity on PPARα, PPARβ/δ, PPARγ
		Currently in phase 2: effect on NAFLD/NASH

Abbreviations: CV, cardiovascular; NAFLD, non-alcoholic fatty liver disease; NASH, non-alcoholic steatohepatitis.

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
