# Peer review of "Peroxisome Proliferator-Activated Receptors and Their Novel Ligands as Candidates for the Treatment of Non-Alcoholic Fatty Liver Disease"

_cells, 2020, doi:10.3390/cells9071638_

Round 1

Reviewer 1 Report

The manuscript by Fougerat and Colleagues deals with the pathomechanisms underlying non-alcoholic fatty liver disease (NAFLD), which represents a major metabolic pathology. Its progression,
complications, and current therapeutic strategies are described. The Authors focus on the roles PPAR isotypes play in the disease. The effects of first-generation PPAR agonists and newly produced and identified molecules acting as ligands are reviewed, in view of their potential use in the treatment of NAFLD.

This review appears exhaustive and timely, for the progressively increasing prevalence of NAFLD, due to the global obesity pandemic, a condition favoring the disease. Classic and updated literature is correctly cited and referred to. This long manuscript is structured in a consequential and logical manner and is overall well written. Carefully conceived and drawn illustrations help the reader following the text and summarizing the reported data, from an original viewpoint. Conclusions are interesting and relevant, even from a therapeutic perspective.

I only have minor comments:

  • In the title, I suggest to avoid unexplained abbreviations (NAFLD)
  • The manuscript is very long and the list of references seems excessive, I would make an effort to synthesize the text and reduce the number of references
  • Few typing errors should be corrected, for example in these references:

235. Kersten, S.; Seydoux, J.; Peters, J.M.; Gonzalez, F.J.; Desvergne, B.; Wahli, W. Peroxisome proliferator–activated receptor $α$ mediates the adaptive response to fasting. J. Clin. Invest. 1999, 103, 1489–1498,
doi:10.1172/JCI6223.

281. Bedu, E.; Wahli, W.; Desvergne, B. Peroxisome proliferator-activated receptor β/δ as a therapeutic target for matabolic diseases. Expert Opin. Ther. Targets 2005, 9, 861–873.

Author Response

We thank the four reviewers for their enthusiasm and constructive evaluation of our manuscript. We appreciate them taking the time to propose ameliorations during these challenging times. Please find our responses to their concerns below. The reviewers’ comments are in black with our specific responses underneath in blue.

Reviewer 1

The manuscript by Fougerat and Colleagues deals with the pathomechanisms underlying non-alcoholic fatty liver disease (NAFLD), which represents a major metabolic pathology. Its progression, complications, and current therapeutic strategies are described. The Authors focus on the roles PPAR isotypes play in the disease. The effects of first-generation PPAR agonists and newly produced and identified molecules acting as ligands are reviewed, in view of their potential use in the treatment of NAFLD.

This review appears exhaustive and timely, for the progressively increasing prevalence of NAFLD, due to the global obesity pandemic, a condition favoring the disease. Classic and updated literature is correctly cited and referred to. This long manuscript is structured in a consequential and logical manner and is overall well written. Carefully conceived and drawn illustrations help the reader following the text and summarizing the reported data, from an original viewpoint. Conclusions are interesting and relevant, even from a therapeutic perspective.

  1. In the title, I suggest to avoid unexplained abbreviations (NAFLD)

Response: We replaced NAFLD by non-alcoholic fatty liver disease in the title (page 1 line 6).

  1. The manuscript is very long and the list of references seems excessive, I would make an effort to synthesize the text and reduce the number of references

Response: Our aim was to write an up to date and comprehensive review of the subject chosen. Short reviews often do not cover a subject completely and it is difficult to find all the relevant information (references) that one wishes to obtain. Yes, a comprehensive review tends to be long and contain lots of references, but we think that this is useful for readers interested to use review articles as important source of information for their work. Therefore, we would like to propose not to shorten the text and reference list. We hope that the reviewer can agree with us that the current manuscript is valuable and interesting enough to justify its length.

  1. Few typing errors should be corrected, for example in these references:

235. Kersten, S.; Seydoux, J.; Peters, J.M.; Gonzalez, F.J.; Desvergne, B.; Wahli, W. Peroxisome proliferator–activated receptor $α$ mediates the adaptive response to fasting. J. Clin. Invest. 1999, 103, 1489–1498,
doi:10.1172/JCI6223.

281. Bedu, E.; Wahli, W.; Desvergne, B. Peroxisome proliferator-activated receptor β/δ as a therapeutic target for matabolic diseases. Expert Opin. Ther. Targets 2005, 9, 861–873.

Response: We apologise for the typing errors in the references, which are now corrected.

Reviewer 2 Report

Manuscript ID: cells-835212
Type of manuscript: Review
Title: Peroxisome proliferator-activated receptors and their novel ligands as candidates for the treatment of NAFLD
Authors: Anne Fougerat *, Alexandra Montagner, Nicolas Loiseau, Hervé Guillou, Walter Wahli *

The Role of PPARs in Disease

The review aimed to describe the determinants and mechanisms underlying the pathogenesis of NAFLD, its progression and complications, as well as the current therapeutic strategies that are employed. The autohrs focused on the complementary and distinct roles of PPAR isotypes in many biological processes and on the effects of first-generation PPAR agonists. They also reviewed novel and safe PPAR agonists with improved efficacy and their potential use in the treatment of NAFLD.

The authors have done an excellent job and I would like to congratulate them for the manuscript. The review describes clearly and deeply the state of the art about NAFLD and PPARs.

I consider the paper is suitable for publication althougt I would like to make some suggestions/comments

  1. To improve the fluidity and make the text easier to read, I should recommend to the authors that they restructure the section 3 of the review. Sections 3.1.2 and 3.1.3 should be combined or more differentiated. Both titles are too similar and confusing. Moreover, I suggest to the authors to make some subsections in these above-mentioned sections.
  2. I miss the full name of some genes/proteins where just the acronym has been included.
  3. Finally, the authors should check the type and size of letters in figure legends as some parts of the text are different.

Author Response

We thank the four reviewers for their enthusiasm and constructive evaluation of our manuscript. We appreciate them taking the time to propose ameliorations during these challenging times. Please find our responses to their concerns below. The reviewers’ comments are in black with our specific responses underneath in blue.

Reviewer 2

The review aimed to describe the determinants and mechanisms underlying the pathogenesis of NAFLD, its progression and complications, as well as the current therapeutic strategies that are employed. The autohrs focused on the complementary and distinct roles of PPAR isotypes in many biological processes and on the effects of first-generation PPAR agonists. They also reviewed novel and safe PPAR agonists with improved efficacy and their potential use in the treatment of NAFLD.

The authors have done an excellent job and I would like to congratulate them for the manuscript. The review describes clearly and deeply the state of the art about NAFLD and PPARs.

  1. To improve the fluidity and make the text easier to read, I should recommend to the authors that they restructure the section 3 of the review. Sections 3.1.2 and 3.1.3 should be combined or more differentiated. Both titles are too similar and confusing. Moreover, I suggest to the authors to make some subsections in these above-mentioned sections.

Response: We agree that the titles of the section 3.1.2 and 3.1.3 are confusing. The title of Section 3.1.2 has been changed to: PPARs in glucose and lipid metabolism (page 17 line 681); the title of Section 3.1.3 has been changed to: PPARs in inflammation and hepatic stellate cell activation (page 20 line 832). We added a Section 3.1.4 with the title: PPARs in NAFLD (page 23 line 930). We preferred not to make an additional subsection level as there are already 2 in this chapter 3 (3.1.1/3.1.2/3.1.3 and now 3.1.4).

  1. I miss the full name of some genes/proteins where just the acronym has been included.

Response: We have added the full name of genes/proteins that were indeed missing.

  1. Finally, the authors should check the type and size of letters in figure legends as some parts of the text are different.

Response: We apologise for these typing errors, which are now corrected in all figure legends.

Reviewer 3 Report

The authors reviewed in a comprehensive way about an interesting topic nowadays. non-alcoholic fatty liver disease (NAFLD) is infact considered as one of the major health issue worldwide. Even if this disease is dramatically increasing in developed countries, there are still no approved therapies. In this review the authors described the multiple factors that contribute to the development of NAFLD nad its progression and underlined the role of the three peroxisome proliferator-activated receptors (PPARs)as promising targets for the treatment of NAFLD.

I consider this review very well structured and able to describe in details each aspect of the disease and of the possible pharmacological strategies available until now. However I'll appreciate if the authors would like to consider some minor suggestions. I think it will be helpful for readers have a scheme or a figure summarizing the similarities in protein structure and mechanism of action of the three PPARs. 

Author Response

We thank the four reviewers for their enthusiasm and constructive evaluation of our manuscript. We appreciate them taking the time to propose ameliorations during these challenging times. Please find our responses to their concerns below. The reviewers’ comments are in black with our specific responses underneath in blue.

Reviewer 3

The authors reviewed in a comprehensive way about an interesting topic nowadays. non-alcoholic fatty liver disease (NAFLD) is infact considered as one of the major health issue worldwide. Even if this disease is dramatically increasing in developed countries, there are still no approved therapies. In this review the authors described the multiple factors that contribute to the development of NAFLD nad its progression and underlined the role of the three peroxisome proliferator-activated receptors (PPARs)as promising targets for the treatment of NAFLD.

I consider this review very well structured and able to describe in details each aspect of the disease and of the possible pharmacological strategies available until now. However I'll appreciate if the authors would like to consider some minor suggestions.

  1. I think it will be helpful for readers have a scheme or a figure summarizing the similarities in protein structure and mechanism of action of the three PPARs. 

Response: We added a figure (Figure 3) to illustrate the similarities in protein structure of PPARs, with information on posttranslational modifications, and their mechanisms of gene transcriptional regulation (page 17).

Reviewer 4 Report

Dear Editor,

the review submitted by Fougerat and colleagues is a package of information about the role of PPARs in the treatment of NAFLD. The review is good prepared and well-written.

I have only a few suggestions from the view of an active scientist in the field of PPAR molecules.

The interplay between PPARs is an issue affecting the assessment and evaluation of biological and clinical outputs in PPARs research world. The authors have already mentioned it in different pieces of review. It would be great if authors can make a small section discussing available information about cross-talk between PPARs in the pathology of the liver. The issue is more important in case of PPARs agonists, it would be informative if authors can add some statements about the effect of an agonist of one PPAR on the other PPAR in the liver.

The review highlighted the current trials using the specific, pan or dual agonists of PPARs. There is an old paradigm dealing with cross-talk between heart and liver. PPAR alpha plays a key role in cardiac physiology and pathology. Is there any information from these trails to show that treating liver patients with PPAR agonist affected heart?

It is often discussed in scientific society that the clinical efforts using PPARs agonists did not work on human. The authors need to argue the potential issues about the not sufficient efficacy of PPAR agonists in clinical trials and discuss alternative strategies to improve it.

Sincerely

Author Response

We thank the four reviewers for their enthusiasm and constructive evaluation of our manuscript. We appreciate them taking the time to propose ameliorations during these challenging times. Please find our responses to their concerns below. The reviewers’ comments are in black with our specific responses underneath in blue.

Reviewer 4

Dear Editor,

the review submitted by Fougerat and colleagues is a package of information about the role of PPARs in the treatment of NAFLD. The review is good prepared and well-written.

I have only a few suggestions from the view of an active scientist in the field of PPAR molecules.

  1. The interplay between PPARs is an issue affecting the assessment and evaluation of biological and clinical outputs in PPARs research world. The authors have already mentioned it in different pieces of review. It would be great if authors can make a small section discussing available information about cross-talk between PPARs in the pathology of the liver. The issue is more important in case of PPARs agonists, it would be informative if authors can add some statements about the effect of an agonist of one PPAR on the other PPAR in the liver.

Response: Although each PPAR isotype is preferentially expressed in certain tissues, though often with varying levels according to the physiological status, the three PPAR isotypes are expressed in several organs. Furthermore, the three PPARs contain a highly conserved DNA-binding domain and bind a same response element (PPRE) in the regulatory regions of target genes. Thus, cross-talks between PPARs in response to a specific isotype agonist may indeed occur in NAFLD. An interplay between PPARα and PPARγ has been reported in brown adipose tissue. A set of genes involved in brown adipose tissue function is activated by both PPARα agonist (fenofibrate) and PPARγ agonist (rosiglitazone) in mice, which suggests a redundancy of the 2 PPARs and may explain why some studies found that PPARα is dispensable for thermogenesis while other findings indicate a role of PPARα in brown adipose tissue function (Shen Y, Cell Reports, 2020). An example of compensation has been observed in PPARα-deficient mice fed a HFD in which hepatic expression of PPARγ and its target genes are highly up-regulated (Patsouris D, Endocrinology, 2006). Moreover, adenoviral overexpression of PPARγ in mouse liver leads to the expression of PPARα target genes involved in fatty acid oxidation and ketogenesis, suggesting that high level of PPARγ expression can compensate for PPARα in the liver (Patsouris D, Endocrinology, 2006). A compensatory role of PPARβ/δ in the repression of hepatic Cyp7b1 in female mice has been shown in the absence of PPARα (Leuenberger N, J Clin Invest, 2009). Altogether, these studies show the existence of cross-talks and compensatory mechanisms between PPAR isotypes which may be important when testing PPAR agonists.

A paragraph addressing this point has been added on page 20 (lines 809-925) of the manuscript.

  1. The review highlighted the current trials using the specific, pan or dual agonists of PPARs. There is an old paradigm dealing with cross-talk between heart and liver. PPAR alpha plays a key role in cardiac physiology and pathology. Is there any information from these trails to show that treating liver patients with PPAR agonist affected heart?

Response: This remark is particularly relevant as PPARα is expressed in the heart where it regulates expression of genes involved in fatty acid uptake and oxidation. Transgenic mice overexpressing PPARα in the heart have increased cardiac fatty acid oxidation and reduced myocardial glucose utilization as it is observed in diabetic heart (Finck BN, J Clin Invest, 2002). However, to our knowledge, there are no studies reporting a deleterious effect of PPARα agonist on cardiac functions in cardiac hypertrophy (Francis GA, Am J Physiol Heart Circ Physiol, 2003). It has been suggested that the extra-cardiac effects of PPARα agonists, by decreasing TG and free fatty acid levels, lead to reduced fatty acid oxidation in the heart by limiting substrate availability for PPARα. The activation of PPARα during cardiac ischemic events is less clear; pretreatment with a PPARα agonist seems to reduce infarct size whereas PPARα activation during the postischemic period inhibits the switch to glucose utilization that is necessary for myocardium repair (Francis GA, Am J Physiol Heart Circ Physiol, 2003).

To date, there is no information in current clinical trials on the direct effects of novel PPAR agonists on the heart. In a posthoc analysis, elafibranor has been shown to not cause cardiovascular events or deaths (Ratziu V, Gastroenterology, 2016). Saroglitazar reduces lipid biomarkers of cardiovascular disease in patients with hypertriglyceridemia (Krishnappa M, Atherosclerosis, 2019) and the cardiovascular risk in diabetic patients (Krishnappa M, Cardiovasc Diabetol, 2020). Pemafibrate is currently being tested for its effect on reducing cardiovascular events in diabetic patients with high TG levels in the PROMINENT study (NCT03071692) (Pradhan AD, Am J Heart, 2018). However, as mentioned above, given that hypertriglyceridemia is associated with the development of cardiovascular disease, PPARα agonists may have beneficial effects on the heart due to their capacity to lower circulating TG  levels.

This information is now given in the concluding remarks (page 32 lines 1322-1329).

  1. It is often discussed in scientific society that the clinical efforts using PPARs agonists did not work on human. The authors need to argue the potential issues about the not sufficient efficacy of PPAR agonists in clinical trials and discuss alternative strategies to improve it.

Response: The differences between mice and human regarding the PPAR is an important issue. For example and as mentioned in our manuscript, hepatic PPARα expression is higher in rodents than in humans. Thus, already in preclinical studies “human-based” strategies are relevant to test drugs targeting PPARs. For example, humanized mice such as transgenic mice carrying humanized PPARs or mice possessing a humanized liver may represent a relevant strategy for the evaluation of PPAR agonists (Boeckmans J, Cells, 2019).

In addition, a better understanding of the molecular mechanisms, especially the coregulator network, underlying PPAR regulation in different species, tissues and diseases may be needed for designing more-specific and more-potent PPAR ligands (Kang Z, FASEB, 2020). PPAR functions are also regulated by several posttranslational modifications that influence protein stability and localization, ligand binding and co-factor interaction. Understanding the role of these posttranslational modifications and their correlation with some diseases might help developing novel molecules such as molecules that specifically inhibit one posttranslational modification (Brunmeir R, IJMS, 2018).

These points are now discussed pages 30 (lines 1259-1269).